# ASYMMETRIC EFFECTS OF SELF-CORRECTIVE LEARNING ON CHAIN-OF-THOUGHT REASONING FOR EFFICIENT POLICY ADAPTATION

## ABSTRACT

Recent advances in language model (LM)-powered agents have demonstrated the potential to tackle complex embodied tasks by grounding the models' commonsense world knowledge in the interactive physical environments in which the agents operate. However, these LM-based agents' adaptation to a stream of diverse tasks over time remains challenging, particularly under limited supervision and resource constraints. In this paper, we present BiCL, an embodied task adaptation framework that addresses the problem of continual LM finetuning across diverse tasks and adaptation stages using only a small dataset per task and a small LM (i.e., with 0.5B parameters). We devise *bidirectional CoT learning*, which jointly optimizes chain-of-thought (CoT) reasoning and reflexive reasoning through per-task bidirectional supervision: few-shot CoT guidance and rationale-wise correction. The latter enables the model to revise its prior rationale trajectories for new tasks, while the former strengthens multi-step task-specific reasoning through minimal demonstrations. This dual optimization allows the agent to adapt more efficiently through forward knowledge transfer over time, ultimately yielding asymmetric effects by fostering robust CoT reasoning at inference without requiring explicit reflection. Furthermore, we implement *rationale-wise test-time scaling*, a mechanism that dynamically adjusts the depth of CoT reasoning based on the model's confidence in actions inferred from its own rationales. Through extensive experiments on VirtualHome and ALFWorld, we demonstrate performance superiority over other LM-based planning and continual task adaptation approaches, while achieving strong efficiency in computation, data usage and model parameters.

## 1 INTRODUCTION

In real-world applications, embodied agents are required to adapt to a stream of tasks over time, as everyday embodied tasks continually shift in surrounding objects, their relations, required skills, and environmental dynamics (Powers et al., 2022; Li et al., 2024a). Recent advances in language model (LM)-powered embodied agents have demonstrated strong capabilities in tackling such open-ended embodied tasks by grounding the commonsense world knowledge encapsulated in pre-trained models to interactive environments (Ahn et al., 2022; Yao et al., 2023; Shinn et al., 2023). Yet, these approaches remain underexplored for efficient task adaptation, particularly in scenarios involving limited data supervision and smaller LMs with restricted reasoning capabilities.

A straightforward solution is to perform either CoT prompting with in-context samples (Huang et al., 2023; Yao et al., 2023) or CoT distillation (i.e., supervised finetuning with rationales) (Choi et al., 2024; DeepSeek-AI et al., 2025) on each discretely incoming stream of task-specific data. However, in practice, limited task-specific supervision, combined with the capacity constraints of small LMs, poses significant challenges for task adaptation. Through our experiments, we observe that in-context CoT prompting approaches (e.g., ReAct (Yao et al., 2023), SayCan (Ahn et al., 2022)) exhibit degraded performance, when applied to small LMs (as shown in Table 1). Furthermore, CoT distillation (e.g., TAIL-Distill (Liu et al., 2024)) yields suboptimal performance under limited data conditions (as shown in Table 2). This highlights the need for effective forward knowledge transfer across adaptation stages, where each stage handles a distinct task, especially in scenarios with limited supervision and small LMs. Lastly, self-correction mechanisms (i.e., refining initially generated

rationales based on feedback) may offer a potential solution to mitigate suboptimal reasoning in small LMs. However, their performance (e.g., Self-Correction (Welleck et al., 2023)) remains limited due to the absence of precise feedback at inference (as shown in Table 3), and they nearly double the inference cost, which is a critical drawback for the practical deployment of embodied agents.

To address this, we introduce the notion of *bidirectional CoT learning*, a simple yet efficient strategy for embodied task adaptation. Specifically, at each task adaptation stage, the learning process involves dual supervision: learning *CoT reasoning* through distillation of multi-step rationales, and learning *reflexive reasoning* through the correction of prior rationales generated by a previously learned policy. This process is efficiently conducted at each stage, using only few-shot demonstrations, specifically task-specific CoT rationales. Reflexive reasoning parallels how humans draw rationales from prior thoughts while actively identifying and correcting discrepancies between earlier reasoning and new contexts, a form of metacognitive behavior that fosters more transferable knowledge in response to evolving scenarios (Yeung & Summerfield, 2012). It extends beyond CoT finetuning by incorporating a reflexive mechanism that delivers richer supervision, allowing agents to internalize refined task-specific knowledge. In this way, *bidirectional CoT learning* produces *asymmetric effects*, reinforcing CoT reasoning via internalized reflexive knowledge and thereby delivering consistently robust performance without explicit reflection at inference.

To this end, we present the BiCL framework, designed for efficient task adaptation in embodied agents through ***bidirectional CoT learning***. At each adaptation stage, we first retrieve the most relevant previously learned policy by comparing the rationales from previous stages with those derived from the current stage's demonstrations. The current policy is initialized with the retrieved prior one and then optimized using bidirectional objectives: generating rationales via CoT reasoning, and correcting rationales generated by the prior policy via reflexive reasoning, both guided by few-shot demonstrations. Following this adaptation, inference is conducted solely through CoT reasoning, without explicitly invoking reflexive reasoning. To improve efficiency, we also devise ***rationale-wise test-time scaling*** which dynamically determines when to terminate CoT reasoning based on the model's confidence in the predicted actions conditioned on generated rationales.

Through evaluation on VirtualHome (Puig et al., 2018) and ALFWorld (Shridhar et al., 2021) benchmarks, we demonstrate the effectiveness of BiCL over existing LM-based planning and continual task adaptation baselines across four aspects. **(i) Continual few-shot adaptation performance:** BiCL achieves robust task success, with an average improvement of 18.54% over the most competitive baseline SeqFT-Distill (Shridhar et al., 2023) on the unseen category in the continual 5-shot adaptation setting (see Table 2). **(ii) Computational efficiency:** BiCL substantially reduces the computation cost of CoT reasoning, lowering the number of generated tokens by $46.01\%$ (see Table 6), while still outperforming the Self-Correction (Welleck et al., 2023), which requires nearly twice as many rationale generation for refinement, with an average improvement of $25.78\%$ (see Table 3). **(iii) Data efficiency:** BiCL demonstrates strong data efficiency, outperforming SeqFT-Distill trained with twice the demonstrations by 7.50% and TAIL-Distill (Liu et al., 2024) by 18.75% on the unseen category (see Table 4). **(iv) Parameter efficiency:** despite using a significantly smaller 0.5B LM, BiCL shows only a moderate performance gap of 9.57% on average compared to a large language model (LLM)-based planner using GPT-4o (Achiam et al., 2023), indicating strong performance efficiency relative to its size. In contrast, other baselines suffer substantially larger degradations (see Table 1). Thanks to these advantages, BiCL effectively enables embodied agents to continuously adapt to a series of new tasks in dynamic environments, even under limited per-task data and resource-constrained settings where only a small LM can be used. The contributions of our work are summarized as follows.

- We present the BiCL framework, designed to address the challenges of embodied task adaptation under constrained data and resource settings.

- We devise *bidirectional CoT learning*, a novel joint training strategy that combines CoT reasoning and reflexive reasoning, supervised respectively by few-shot CoT guidance and rationale-wise correction of prior knowledge. Notably, this yields asymmetric effects, strengthening CoT reasoning at inference without requiring explicit reflection.

- We introduce *rationale-wise test-time scaling*, by which the depth of CoT reasoning is dynamically adjusted. This mechanism improves computational efficiency and enhances task success, thereby removing the need for explicit self-correction.

- Through extensive experiments on VirtualHome and ALFWorld benchmarks, we demonstrate both the performance superiority and data efficiency of BiCL in task adaptation.

## 2 RELATED WORK

Embodied task planning has gained significant attention, driven by the advancements in LM's reasoning capabilities (Jiang et al., 2022; Driess et al., 2023; Huang et al., 2022a; 2023; Ahn et al., 2022). In parallel, several works have focused on leveraging a stream of datasets to progressively learn diverse tasks over time (Liu et al., 2024; Schmied et al., 2023; Kim et al., 2024) or adopt RL-based learning schemes to acquire task-specific behaviors through online interaction (Feng et al., 2025; Chen et al., 2022; Ye et al., 2024). Recently, a growing body of work has explored distilling such reasoning capabilities from LLMs into smaller models (Choi et al., 2024; Li et al., 2023; Shridhar et al., 2023; DeepSeek-AI et al., 2025). Building on this line of research, our work seeks to equip LM-based agents with robust CoT reasoning through bidirectional CoT learning, enabling effective adaptation to a stream of new tasks under limited supervision and model capacity. Further details on related work are provided in Appendix A.

## 3 PRELIMINARIES

### 3.1 PROBLEM FORMULATION

We consider a stream of few-shot demonstrations $\{\mathcal{D}_1, \mathcal{D}_2, ..., \mathcal{D}_H\}$, where $i$-th stage demonstrations $\mathcal{D}_i$ contain expert transitions. Each transition $d = (\mathcal{T}, o, a, \mathcal{Z})$ is represented as a tuple of task $\mathcal{T}$, observation $o$, action (plan) $a$, and a rationale set $\mathcal{Z}$. A task is defined by an underlying instructional template in language, serving as a generalizable form. Similar to the formulation in (Kim et al., 2024), each task either allows linguistic or object-level variations while maintaining consistent behavioral semantics, or accommodates behavioral variations under the same environmental conditions. For example, the $i$-th stage task $\mathcal{T}_i$ might include instructions such as "*clean an apple and put in the cabinet*", "*clean a towel and put in the washing machine*", and other variations that share similar behavioral patterns, which the agent is expected to follow. The observation captures the currently perceptible information from the environment, rendering the partially-observable nature of embodied task planning. The observation is represented as a set of triplets, each consisting of a source entity, a relation, and a target entity, e.g., *(apple, on, table)*. The rationale set $\mathcal{Z} = \{z_k\}_{k=1}^N$ contains key elements essential for embodied task planning (Yao et al., 2023; Choi et al., 2024), such as target object locations and sub-goals.

Formally, let $\{\mathcal{T}_1, \mathcal{T}_2, \ldots, \mathcal{T}_H\}$ be the sequence of adaptation tasks and $\mathbf{SR}(\pi, \mathcal{T}_i) \in [0, 1]$ the task success rate of policy $\pi$ on task $\mathcal{T}_i$. The objective of continual adaptation is to find

$$\pi^* = \arg\max_\pi \sum_{i=1}^{H} \mathbf{SR}(\pi, \mathcal{T}_i). \tag{1}$$

i.e., the policy that maximizes the cumulative task success rate across all $H$ stages.

### 3.2 CONTINUAL TASK ADAPTATION WITH PRE-TRAINED MODELS

To continuously adapt to tasks while leveraging the knowledge embedded in pre-trained models, recent works adopt parameter-efficient tuning modules in LMs (Liu et al., 2024; Schmied et al., 2023). Following this approach, we implement an embodied agent policy using Low-Rank Adaptation (LoRA) (Hu et al., 2022), where trainable adapter parameters $\theta$ are integrated with frozen LM parameters $\theta_{\mathrm{LM}}$, i.e., $\pi(\cdot; \theta_{\mathrm{LM}}, \theta)$. For simplicity, we use the notation $\pi(\cdot; \theta)$, omitting $\theta_{\mathrm{LM}}$. Furthermore, we structure the embodied agent using separate modules for reasoning and planning, corresponding to a reasoning-policy and a planning-policy, each formed by distinct adapters $\theta_z$ and $\theta_p$, respectively. The reasoning-policy is responsible for rationale generation, while the planning-policy produces actions based on the generated rationales.

## 4 BiCL FRAMEWORK

To address the challenge of embodied task adaptation across sequential stages, we present the BiCL framework comprising of (i) *bidirectional CoT learning* and (ii) *rationale-wise test-time scaling*, as depicted in Figure 1, where each stage receives few-shot demonstrations $\mathcal{D}$. (i) For adaptation,

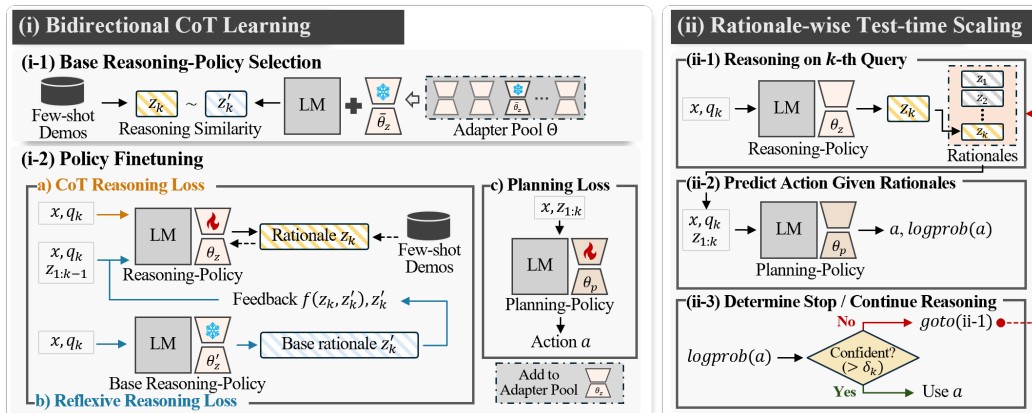

Figure 1: BiCL framework: In (i-1), the most relevant reasoning-policy is selected from the adapter pool to serve as the base one. In (i-2), the reasoning-policy is then finetuned through bidirectional joint optimization on CoT reasoning and reflexive reasoning losses. (a) The former loss is supervised by multi-step rationales in few-shot demonstrations, and (b) the latter loss is formalized as the correction of base rationales generated by the base reasoning-policy. In (ii), the CoT reasoning depth is dynamically adjusted based on the planning-policy's confidence in predicted actions.

we first select a base reasoning-policy by evaluating the similarity between rationales generated by previously learned policies and those in the current stage's demonstrations. The selected one is used to initialize the reasoning-policy for the current adaptation stage. Then, the reasoning-policy is finetuned through jointly optimizing CoT reasoning and reflexive reasoning objectives. CoT reasoning is trained using rationales derived from demonstrations, while reflexive reasoning is formulated as the correction of base rationales generated by the selected base reasoning-policy, guided by feedback. The planning-policy predicts actions, conditioned on incrementally accumulated rationales. (ii) For inference, we dynamically scale the CoT reasoning process based on the planning-policy's confidence. The following two subsections describe how these two procedures are performed within a single stage with demonstrations $\mathcal{D}_i$ to enable task adaptation, where the stage notation $i$ is omitted for simplicity.

## 4.1 BIDIRECTIONAL CoT LEARNING

**Base reasoning-policy selection.** At the beginning of each adaptation stage, we select a reasoning-policy $\pi_z$ learned from the previous stage to serve as the base reasoning-policy. Such base reasoning-policy is chosen by evaluating the similarity between rationales $z_k \in \mathcal{Z}$ in the demonstrations and the rationales $\bar{z}_k$ generated by each candidate reasoning-policy formed by an adapter $\bar{\theta}_z \in \Theta$ in the adapter pool $\Theta$. Note that the adapter pool contains the adapters learned up to the current stage. Then, the reasoning-policy maximizing the sentence embedding similarity SIM is chosen as

$$\theta_z' = \arg\max_{\bar{\theta}_z \in \Theta} \sum_{(x, \mathcal{Z}) \in \mathcal{D}} \sum_{k=1}^{N} \text{SIM}\,(z_k, \bar{z}_k) \text{ where } \bar{z}_k \sim \pi_z(\cdot | x, q_k; \bar{\theta}_z). \tag{2}$$

Here and in what follows, $x = (\mathcal{T}, o, h)$ denotes a triple of task $\mathcal{T}$, observation $o$, and action history $h$, and $q_k$ is the query for $k$-th rationale generation. The base reasoning-policy $\pi_z(\cdot; \theta_z')$ is used to generate base rationales for subsequent correction and to initialize the adapter for the current stage.

**CoT reasoning loss.** To equip the reasoning-policy with CoT reasoning capabilities, we train it to generate rationales segment by segment. The loss for CoT reasoning is then defined as

$$\mathcal{L}_{\text{CoT}}(\theta_z) = \mathop{\mathbb{E}}_{(x, \mathcal{Z}) \sim \mathcal{D}} \left[ \sum_{k=1}^{N} - \log \pi_z(z_k | x, q_k; \theta_z) \right]. \tag{3}$$

**Reflexive reasoning loss.** In addition to CoT reasoning, the reasoning-policy is trained to incrementally correct base rationales $z_k' \sim \pi_z(\cdot | x, q_k; \theta_z')$ using feedback, focusing on one specific rationale at

---

**Algorithm 1** BiCL framework: bidirectional CoT learning

---

1: **Input:** a stream of few-shot demonstrations $\{\mathcal{D}_1, \mathcal{D}_2, ..., \mathcal{D}_H\}$
2: Initialize adapter pool $\Theta = \emptyset$
3: **for** each adaptation stage $i \leftarrow 1, ..., H$ **do**
4:     Select adapter from adapter pool for base reasoning-policy $\pi_z(\cdot; \theta'_z)$ using equation 2
5:     Initialize adapters for the current stage $\theta_z \leftarrow \theta'_z, \theta_p$
6:     **while** not converged **do**
7:         Sample a batch of $\{(\mathcal{T}, o, a, \mathcal{Z})\} \sim \mathcal{D}_i$
8:         Generate base rationales $z'_k \sim \pi_z(\cdot|x, q_k; \theta'_z)$ through the base reasoning-policy
9:         Update reasoning-policy $\pi_z(\cdot; \theta_z)$ using loss $\mathcal{L}_{\text{reasoning}}$ in equation 5
10:        Update planning-policy $\pi_p(\cdot; \theta_p)$ using loss $\mathcal{L}_{\text{planning}}$ in equation 6
11:     Add adapters to adapter pool $\Theta \leftarrow \Theta \cup \{\theta_z\}$

---

each correction step. The feedback $f(z_k, z'_k)$ is provided in natural language, categorized as "Major revision", "Moderate revision", and "Minor revision". This categorization is determined based on the sentence similarity $\text{SIM}(z_k, z'_k)$ between the base rationale $z'_k$ and its corresponding rationale $z_k$ from the demonstrations. Then, the loss for reflexive reasoning is defined as

$$\mathcal{L}_{\text{reflexive}}(\theta_z) = \mathop{\mathbb{E}}_{(x,\mathcal{Z})\sim\mathcal{D}} \left[ \sum_{k=1}^{N} -\log \pi_z(z_k|z'_k, x, q_k, f(z_k, z'_k), z_{1:k-1}; \theta_z) \right] \tag{4}$$

where $z_{i:j} = \{z_i, \ldots, z_j\}$ denotes the sequence of previous rationales, and $z_{j:k} = \emptyset$ if $j > k$.

This *bidirectional CoT learning* strategy extends beyond mere CoT learning by systematically incorporating feedback-driven adjustment into reflexive reasoning steps, which identify discrepancies in prior knowledge. This enables the agent to further enhance task-specific knowledge, effectively improving its CoT reasoning capabilities. Accordingly, the final loss for the reasoning-policy $\pi_z$ is defined as

$$\mathcal{L}_{\text{reasoning}}(\theta_z) = \mathcal{L}_{\text{CoT}}(\theta_z) + \mathcal{L}_{\text{reflexive}}(\theta_z). \tag{5}$$

**Planning loss.** To predict actions based on the rationales, we optimize the planning-policy $\pi_p$ via an action reconstruction loss, where the policy is incrementally conditioned on the first $k$ rationales. This encourages the planning-policy to integrate partial reasoning signals. The loss is defined as

$$\mathcal{L}_{\text{planning}}(\theta_p) = \mathop{\mathbb{E}}_{(x,a,\mathcal{Z})\sim\mathcal{D}} \left[ \sum_{k=1}^{N} -\log \pi_p(a|x, z_{1:k}; \theta_p) \right] \tag{6}$$

where $x = (\mathcal{T}, o, h)$. At the end of the adaptation, the adapter $\theta_z$ is added to the adapter pool $\Theta$ for use in subsequent adaptation stages. Algorithm 1 lists the adaptation procedures of BiCL.

## 4.2 Rationale-wise test-time scaling

As the full set of rationales may be unnecessary for action prediction, depending on task complexity (Yao et al., 2023), we devise a test-time scaling mechanism in which the CoT reasoning depth is dynamically adjusted. At each CoT reasoning step $k$, the planning policy's confidence in its predicted action, conditioned on the first $k$ rationales $z_{1:k}$, is evaluated to determine whether to terminate or proceed to the next CoT reasoning step. Specifically, if the confidence on the predicted action does not exceed a predefined threshold $\delta_k$ at $k$-th reasoning step, an additional rationale $z_{k+1}$ is generated; otherwise, the corresponding action $a$ is used. Thus, *rationale-wise test-time scaling* is performed as

$$\text{TTS}(k) = \begin{cases} \text{generate next rationale } z_{k+1} & \text{if } \log \pi_p(a|x, z_{1:k}; \theta_p)/|a| \leq \delta_k \\ \text{use } a & \text{otherwise} \end{cases} \tag{7}$$

where $|a|$ is the token length of action $a$. The threshold is computed for each $k$ using the demonstrations by measuring the mean log-probability of the ground-truth action when the trained planning-policy is conditioned on the rationales $z_{1:k}$. This allows for efficient use of computational resources by adaptively scaling the CoT reasoning depth. The complete procedure for rationale-wise test-time scaling in BiCL is provided in Algorithm 3 in Appendix C.9.

## 5 EXPERIMENTS

### 5.1 EXPERIMENT SETTINGS

**Environments.** For evaluation, we use VirtualHome (Puig et al., 2018) and ALFWorld (Shridhar et al., 2021). To configure the continual task adaptation setups, we follow the approach outlined in Kim et al. (2024). Specifically, in **Behavior Incremental Learning (Beh-IL)**, the agent is tasked to incrementally learn new behaviors, while in **Environment Incremental Learning (Env-IL)**, the agent is tasked to incrementally learn to perform behaviors in novel indoor scenes. We evaluate performance on two task categories. In the **Seen** category, instructions and indoor scenes are identical to those in the demonstrations, with variations only in the initial positions of objects. In the **Unseen** category, both instructions and indoor scenes are varied. Further details are in Appendix B.

**Datasets.** Our default few-shot setting uses 20 expert demonstrations per adaptation stage, while more constrained scenarios use only 5 demonstrations. The rationales in these demonstrations can be either annotated by humans or generated by language models; for annotation consistency and quality in datasets, we rely on GPT-4o-mini (Achiam et al., 2023) in our implementation.

**Evaluation metrics.** We use two metrics from Shridhar et al. (2020) to evaluate performance for embodied task planning. **Success rate (SR)** measures the proportion of episodes the agent completes successfully. **Goal success rate (GC)** measures the proportion of satisfied sub-goals out of all given goals. For both, we report the average performance achieved by the agent after each adaptation stage.

**Baselines.** We compare with several LM-based planning and continual task adaptation baselines. **(i) ReAct** (Yao et al., 2023) interleaves rationale generation and action prediction for embodied planning. **(ii) SayCan** (Ahn et al., 2022) integrates action feasibility into ReAct. **(iii) TAIL-Distill** (Liu et al., 2024) employs task-specific adapters for each adaptation stage. We implement two variants for comparison: **TAIL-Action**, which is trained only on state-action pairs from demonstrations; **TAIL-Distill**, which additionally incorporates rationales as BiCL. **(iv) SeqFT-Distill** (Shridhar et al., 2023) uses CoT reasoning to distill step-by-step rationales, with each distilled policy for the current stages initializing the next stage. **(v) CAMA-Distill** (Kim et al., 2024) replays action logits from previous stages and dynamically adjusts their update weights and fine-tunes with rationales. Note that ReAct and SayCan use the same rationale-annotated demonstrations as BiCL, but as in-context examples, while the other baselines (except for TAIL-Action) exploit these demonstrations for CoT reasoning distillation. At inference, all methods rely solely on CoT reasoning **without explicit reflection**. Additionally, we compare against **(vi) LLM-Planner** (Huang et al., 2023), which uses LLMs (e.g. GPT-4o), prompted with expert demonstrations as in-context examples.

**Implementation details.** For finetuning LM-based policies, we employ Qwen2.5-0.5B (Yang et al., 2024) with LoRA adapters (Hu et al., 2022). We also use a language embedding model of paraphrase-MiniLM-L6-v2 (Reimers & Gurevych, 2019) and TF-IDF score (Ramos et al., 2003) to compute the sentence similarity for base reasoning-policy selection and feedback provision.

### 5.2 MAIN RESULTS

Table 1 shows the continual few-shot adaptation performance of BiCL and the baselines under two different setups (Beh-IL and Env-IL) in VirtualHome and ALFWorld. Here, the continual adaptation stage is set to 4. BiCL w/o TTS refers to BiCL without *rationale-wise test-time scaling* in Section 4.2. As shown, BiCL consistently yields robust performance across all cases. In the seen category, it achieves average improvements of 18.39% in SR and 11.65% in GC over the most competitive baseline SeqFT-Distill, and in the unseen category, BiCL outperforms SeqFT-Distill by 17.36% in SR and 12.32% in GC on average. When compared to LLM-Planner utilizing GPT-4o, BiCL exhibits a modest performance gap, 9.57% in SR and 9.59% in GC, despite only using a 0.5B-parameter model. In contrast, SeqFT-Distill suffers substantially larger drops of 27.45% in SR and 21.58% in GC. These results highlight the strong efficiency of BiCL under limited model capacity.

In addition, we validate the effectiveness of *rationale-wise test-time scaling*, as BiCL achieves slightly higher performance over its variant (BiCL w/o TTS), with an average SR increase of 6.68% in the seen category and 6.05% in the unseen. We conjecture that due to limited data at each stage, certain rationales might be under-optimized. Thus, by allowing the model to terminate reasoning early when confidence is sufficiently high, the agent can make decisions more reliably.

Table 1: Continual 20-shot adaptation performance in VirtualHome and ALFWorld: The baselines and BiCL are evaluated in two setups (Beh-IL and Env-IL). The continual adaptation is structured over four stages, where each has 20 demos. The SR and GC are reported in $95\%$ confidence intervals. The best performance in each case among small LM-based approaches is highlighted in **bold**. Except for LLM-Planner employing GPT-4o, all methods use the same small LM of Qwen2.5-0.5B.

| Method | Behavior Incremental Learning (Beh-IL) | | | | Environment Incremental Learning (Env-IL) | | | |
| --- | --- | --- | --- | --- | --- | --- | --- | --- |
| | Seen | | Unseen | | Seen | | Unseen | |
| | SR (%) | GC (%) | SR (%) | GC (%) | SR (%) | GC (%) | SR (%) | GC (%) |
| **Benchmark: VirtualHome** | | | | | | | | |
| LLM-Planner (GPT-4o) | $81.63_{\pm 1.63}$ | $91.79_{\pm 0.87}$ | $76.44_{\pm 1.90}$ | $90.11_{\pm 0.99}$ | $88.38_{\pm 1.43}$ | $92.88_{\pm 0.80}$ | $70.26_{\pm 2.14}$ | $80.31_{\pm 1.51}$ |
| ReAct | $12.06_{\pm 1.40}$ | $30.10_{\pm 1.58}$ | $8.65_{\pm 0.00}$ | $26.31_{\pm 1.38}$ | $20.07_{\pm 1.89}$ | $40.46_{\pm 1.67}$ | $13.96_{\pm 1.67}$ | $33.23_{\pm 1.63}$ |
| SayCan | $46.50_{\pm 2.16}$ | $65.25_{\pm 1.62}$ | $37.96_{\pm 2.04}$ | $56.76_{\pm 1.70}$ | $60.00_{\pm 1.89}$ | $73.99_{\pm 1.37}$ | $48.64_{\pm 1.50}$ | $67.11_{\pm 1.08}$ |
| TAIL-Action | $55.00_{\pm 2.29}$ | $69.58_{\pm 1.74}$ | $33.45_{\pm 2.22}$ | $52.45_{\pm 1.94}$ | $61.56_{\pm 1.85}$ | $78.24_{\pm 1.23}$ | $36.21_{\pm 1.42}$ | $58.93_{\pm 1.11}$ |
| TAIL-Distill | $60.25_{\pm 2.20}$ | $73.44_{\pm 1.66}$ | $46.29_{\pm 2.25}$ | $62.68_{\pm 1.90}$ | $71.88_{\pm 1.70}$ | $83.98_{\pm 1.11}$ | $43.78_{\pm 1.45}$ | $63.91_{\pm 1.10}$ |
| SeqFT-Distill | $65.75_{\pm 2.20}$ | $79.09_{\pm 1.52}$ | $48.68_{\pm 2.30}$ | $64.11_{\pm 1.92}$ | $78.13_{\pm 1.53}$ | $87.66_{\pm 1.00}$ | $57.89_{\pm 1.43}$ | $73.04_{\pm 1.43}$ |
| CAMA-Distill | $59.13_{\pm 2.22}$ | $71.28_{\pm 1.74}$ | $44.28_{\pm 2.28}$ | $59.60_{\pm 1.96}$ | $77.81_{\pm 1.56}$ | $87.69_{\pm 0.96}$ | $43.46_{\pm 1.45}$ | $63.27_{\pm 1.10}$ |
| **BiCL w/o TTS** | $76.13_{\pm 1.91}$ | $83.87_{\pm 1.39}$ | $60.29_{\pm 2.32}$ | $69.13_{\pm 1.97}$ | $85.94_{\pm 1.29}$ | $91.56_{\pm 0.84}$ | $62.81_{\pm 2.58}$ | $75.37_{\pm 1.02}$ |
| **BiCL** | $\mathbf{81.38}_{\pm 1.74}$ | $\mathbf{85.81}_{\pm 1.36}$ | $\mathbf{64.03}_{\pm 2.29}$ | $\mathbf{71.60}_{\pm 1.94}$ | $\mathbf{94.06}_{\pm 0.76}$ | $\mathbf{96.13}_{\pm 0.52}$ | $\mathbf{71.99}_{\pm 1.27}$ | $\mathbf{80.56}_{\pm 0.97}$ |
| **Benchmark: ALFWorld** | | | | | | | | |
| LLM-Planner (GPT-4o) | $91.75_{\pm 0.98}$ | $94.63_{\pm 0.71}$ | $86.75_{\pm 1.83}$ | $88.88_{\pm 1.50}$ | $92.58_{\pm 1.07}$ | $93.95_{\pm 0.81}$ | $95.63_{\pm 0.92}$ | $95.78_{\pm 0.68}$ |
| ReAct | $0.00_{\pm 0.00}$ | $4.34_{\pm 0.36}$ | $0.00_{\pm 0.00}$ | $2.24_{\pm 0.21}$ | $0.00_{\pm 0.00}$ | $1.07_{\pm 0.19}$ | $0.00_{\pm 0.00}$ | $0.70_{\pm 0.13}$ |
| SayCan | $22.81_{\pm 1.15}$ | $47.89_{\pm 1.41}$ | $0.63_{\pm 0.21}$ | $23.28_{\pm 1.53}$ | $21.88_{\pm 1.40}$ | $44.08_{\pm 1.67}$ | $4.38_{\pm 0.00}$ | $22.50_{\pm 1.59}$ |
| TAIL-Distill | $53.91_{\pm 1.65}$ | $59.69_{\pm 1.44}$ | $39.53_{\pm 2.16}$ | $46.95_{\pm 2.01}$ | $47.56_{\pm 2.13}$ | $68.13_{\pm 1.64}$ | $31.41_{\pm 2.45}$ | $46.40_{\pm 2.17}$ |
| TAIL-Action | $56.11_{\pm 1.78}$ | $68.71_{\pm 1.44}$ | $37.97_{\pm 2.50}$ | $55.72_{\pm 2.12}$ | $47.56_{\pm 2.13}$ | $63.23_{\pm 1.69}$ | $24.63_{\pm 2.24}$ | $43.06_{\pm 2.05}$ |
| SeqFT-Distill | $64.30_{\pm 1.70}$ | $71.85_{\pm 1.44}$ | $47.97_{\pm 2.42}$ | $57.98_{\pm 2.16}$ | $61.91_{\pm 2.06}$ | $70.71_{\pm 1.72}$ | $39.22_{\pm 2.47}$ | $51.24_{\pm 2.23}$ |
| CAMA-Distill | $53.20_{\pm 1.64}$ | $59.05_{\pm 1.48}$ | $38.59_{\pm 2.27}$ | $45.95_{\pm 2.11}$ | $55.76_{\pm 2.08}$ | $65.37_{\pm 1.70}$ | $34.38_{\pm 2.46}$ | $46.05_{\pm 2.24}$ |
| **BiCL w/o TTS** | $81.64_{\pm 1.46}$ | $85.21_{\pm 1.24}$ | $64.84_{\pm 2.52}$ | $72.67_{\pm 2.34}$ | $73.24_{\pm 1.80}$ | $78.87_{\pm 1.45}$ | $51.09_{\pm 2.58}$ | $63.19_{\pm 2.21}$ |
| **BiCL** | $\mathbf{85.70}_{\pm 1.34}$ | $\mathbf{88.37}_{\pm 1.15}$ | $\mathbf{70.78}_{\pm 2.41}$ | $\mathbf{77.38}_{\pm 1.99}$ | $\mathbf{82.52}_{\pm 1.57}$ | $\mathbf{85.61}_{\pm 1.28}$ | $\mathbf{56.41}_{\pm 2.56}$ | $\mathbf{66.12}_{\pm 2.14}$ |

Table 2: Continual 5-shot adaptation performance

| Method | VirtualHome Beh-IL | | | | ALFWorld Env-IL | | | |
| --- | --- | --- | --- | --- | --- | --- | --- | --- |
| | Seen | | Unseen | | Seen | | Unseen | |
| | SR (%) | GC (%) | SR (%) | GC (%) | SR (%) | GC (%) | SR (%) | GC (%) |
| TAIL-Distill | $53.87_{\pm 2.11}$ | $69.88_{\pm 1.62}$ | $38.54_{\pm 2.12}$ | $58.45_{\pm 1.81}$ | $33.57_{\pm 1.81}$ | $49.18_{\pm 1.62}$ | $29.64_{\pm 2.06}$ | $45.40_{\pm 1.95}$ |
| SeqFT-Distill | $49.96_{\pm 2.21}$ | $63.99_{\pm 1.71}$ | $41.35_{\pm 2.17}$ | $62.38_{\pm 1.75}$ | $35.54_{\pm 1.86}$ | $51.93_{\pm 1.67}$ | $28.66_{\pm 2.20}$ | $45.34_{\pm 2.04}$ |
| CAMA-Distill | $45.65_{\pm 2.19}$ | $65.27_{\pm 2.12}$ | $36.42_{\pm 2.12}$ | $57.86_{\pm 1.79}$ | $29.55_{\pm 1.57}$ | $44.17_{\pm 1.46}$ | $21.61_{\pm 1.68}$ | $38.30_{\pm 1.72}$ |
| **BiCL w/o TTS** | $66.33_{\pm 2.14}$ | $77.77_{\pm 1.57}$ | $52.27_{\pm 2.25}$ | $66.33_{\pm 1.92}$ | $57.86_{\pm 2.01}$ | $68.42_{\pm 1.67}$ | $48.21_{\pm 2.58}$ | $60.10_{\pm 2.16}$ |
| **BiCL** | $\mathbf{73.68}_{\pm 1.92}$ | $\mathbf{80.99}_{\pm 1.47}$ | $\mathbf{54.56}_{\pm 2.29}$ | $\mathbf{67.38}_{\pm 1.97}$ | $\mathbf{68.48}_{\pm 1.89}$ | $\mathbf{76.09}_{\pm 1.52}$ | $\mathbf{53.66}_{\pm 2.55}$ | $\mathbf{64.20}_{\pm 2.12}$ |

In these experiments, ReAct exhibits the lowest performance mainly due to the limited reasoning capabilities of the small LMs it relies on. SayCan achieves better performance by integrating action feasibility into embodied planning. In contrast, TAIL-Distill and SeqFT-Distill leverage LLM-generated rationales to distill CoT reasoning capabilities, resulting in improved planning performance. While CAMA-Distill preserves knowledge acquired from previous stages via logit replay, it still underperforms compared to BiCL due to a lack of effective reasoning knowledge transfer across stages. BiCL effectively transfers reasoning knowledge through *bidirectional CoT learning*, which strengthens CoT reasoning by correcting the prior knowledge in response to novel tasks.

In Table 2, we evaluate the performance under a more constrained few-shot setting, using 5 demonstrations per stage. Here, the continual adaptation stage is set to 7. As shown, BiCL outperforms SeqFT-Distill by 29.12% higher SR and 19.5% higher GC in the seen category and 18.54% higher SR and 12.14% higher GC in the unseen. This result further highlights the superiority of BiCL in scenarios with severely limited supervision.

## 5.3 ANALYSIS AND ABLATION STUDIES

**Does BiCL enable computationally efficient yet effective inference?** In Table 3, we evaluate *w/ self-correct*, a variant of BiCL that performs explicit reflection on the initially generated rationales at inference. We also compare against Self-Correction (Welleck et al., 2023), which is trained to

correct its *own* generated rationales (unlike BiCL, which is trained to correct rationales produced by *prior* policies). As shown, explicit reflection yields modest gains over BiCL w/o TTS, with a 2.19% SR increase in the seen category and 1.86% in the unseen, but at the cost of generating twice as many rationales. BiCL achieves superior performance across all cases, outperforming Self-Correction by 25.78% SR in the seen category and 22.74% in the unseen. The limited effectiveness of explicit correction is attributed to the lack of precise feedback available at inference time, and the results highlight that bidirectional CoT learning offers greater benefits than correcting a model's own rationales. Moreover, BiCL delivers clear computational efficiency, reducing rationale generation to one-quarter of that required by Self-Correction. This efficiency gain arises because BiCL eliminates the need for explicit reflection and dynamically adjusts the depth of CoT reasoning.

Table 3: Computational efficiency in BiCL measured by SR (%)

| Method | ALFWorld Beh-IL | | ALFWorld Env-IL | | Generated Rationales |
| | Seen | Unseen | Seen | Unseen | |
|---|---|---|---|---|---|
| **BiCL** | **85.70**$_{\pm1.34}$ | **70.78**$_{\pm2.41}$ | **82.52**$_{\pm1.57}$ | **56.41**$_{\pm2.56}$ | 49% |
| w/o TTS | 81.64$_{\pm1.46}$ (-4.06) | 64.84$_{\pm2.52}$ (-5.59) | 73.24$_{\pm1.80}$ (-9.28) | 51.09$_{\pm2.58}$ (-5.32) | 100% |
| w/o TTS & w/ *self-correct* | 83.83$_{\pm1.36}$ (-1.87) | 66.09$_{\pm2.51}$ (-4.69) | 75.10$_{\pm1.76}$ (-7.42) | 53.59$_{\pm2.59}$ (-2.82) | 200% |
| Self-Correction | 56.48$_{\pm1.68}$ (-29.22) | 44.06$_{\pm2.31}$ (-26.72) | 63.18$_{\pm2.03}$ (-22.34) | 37.66$_{\pm2.53}$ (-18.75) | 200% |

**Does BiCL enable data-efficient adaptation?** In Table 4, we compare against SeqFT-Distill and TAIL-Distill, which is most competitive comparisons, under two data augmentation settings in ALFWorld. 2× denotes training with twice the number of expert demonstrations, each augmented with a single set of rationales. *aug* denotes using the original number of expert demonstrations, but augmenting each demonstration with two sets of rationales. Note that BiCL is trained with the original number of expert demonstrations with a single set of rationales. As shown, despite using less reasoning data, BiCL outperforms the augmentation-based variants: it improves SR by 14.84% and 15.47% on the seen and unseen categories of ALFWorld-Beh-IL, and by 17.87% and 15.47% on those of ALFWorld-Env-IL, respectively, over the SeqFT-Distill (*aug*) setting. Moreover, relative to the SeqFT-Distill (2×), BiCL achieves a slight average SR improvement of 8.79% for the seen and 7.50% for the unseen category, despite using only half the data. This strong data efficiency of BiCL is attributed to the *bidirectional CoT learning* strategy which effectively transfers prior knowledge.

Table 4: Data-efficient adaptation in BiCL measured by SR (%)

| Method | ALFWorld Beh-IL | | ALFWorld Env-IL | |
| | Seen | Unseen | Seen | Unseen |
|---|---|---|---|---|
| **BiCL** | **85.70**$_{\pm1.34}$ | **70.78**$_{\pm2.41}$ | **82.52**$_{\pm1.57}$ | **56.41**$_{\pm2.56}$ |
| SeqFT-Distill (2×) | 78.67$_{\pm1.43}$ (-7.03) | 63.28$_{\pm2.49}$ (-7.50) | 71.97$_{\pm1.79}$ (-10.55) | 48.91$_{\pm2.62}$ (-7.50) |
| TAIL-Distill (2×) | 66.56$_{\pm1.56}$ (-19.14) | 50.16$_{\pm2.37}$ (-20.62) | 61.72$_{\pm1.95}$ (-20.80) | 39.53$_{\pm2.49}$ (-16.88) |
| SeqFT-Distill (*aug*) | 70.86$_{\pm1.63}$ (-14.84) | 55.31$_{\pm2.54}$ (-15.47) | 64.65$_{\pm1.96}$ (-17.87) | 40.94$_{\pm2.60}$ (-15.47) |
| TAIL-Distill (*aug*) | 62.19$_{\pm1.73}$ (-23.51) | 48.28$_{\pm2.37}$ (-22.50) | 61.04$_{\pm1.98}$ (-21.48) | 40.63$_{\pm2.48}$ (-15.78) |

**Does reflexive reasoning contribute to forward transfer?** In Table 5, we evaluate two ablated variants of BiCL in ALFWorld. *w/o reflexive* uses the base reasoning-policy as the initialization for the current stage's policy, but is optimized without the reflexive reasoning loss in equation 4. *w/o base* does not leverage the base reasoning-policy in any form. BiCL, with the reflexive reasoning loss, achieves relative gains in SR of 15.78 and 16.87% for the seen and unseen categories of ALFWorld Beh-IL, and 17.36 and 11.25% on those of ALFWorld Env-IL, respectively, over the *w/o reflexive* variant. Furthermore, ablating the base reasoning-policy initialization (i.e., the *w/o base* variant) incurs, on average, an additional performance degradation in SR of 6.73% for the seen and 8.83% for the unseen category relative to *w/o reflexive*. This reveals that BiCL effectively leverages knowledge acquired from prior tasks through reflexive reasoning and base reasoning-policy selection.

**How does *rationale-wise test-time scaling* adapt to the complexity of an instruction?** In Table 6, we report the average plan length required for task completion and the percentage of the rationale $z_k$ at which reasoning is terminated. We also report the percentage of reduction in reasoning tokens. For this, we analyze instruction templates in VirtualHome. As shown, for tasks requiring shorter plans (e.g., TURNON and OPEN), the reasoning process often stops at earlier rationales. In contrast, more complex tasks (e.g., PUTIN and PLACEON) tend to require late-stage rationales such as $z_4$ and $z_5$,

Table 5: Forward knowledge transfer in BiCL measured by SR (%)

| Method | ALFWorld Beh-IL | | ALFWorld Env-IL | |
|---|---|---|---|---|
| | Seen | Unseen | Seen | Unseen |
| **BiCL** | $85.70_{\pm1.34}$ | $\mathbf{70.78}_{\pm2.41}$ | $82.52_{\pm1.57}$ | $\mathbf{56.41}_{\pm2.56}$ |
| *w/o reflexive* | $69.92_{\pm1.64}$ (-15.78) | $53.91_{\pm2.64}$ (-16.87) | $65.16_{\pm1.96}$ (-17.36) | $45.16_{\pm2.61}$ (-11.25) |
| *w/o base* | $65.16_{\pm1.72}$ (-20.54) | $47.66_{\pm2.58}$ (-23.12) | $56.47_{\pm2.02}$ (-26.05) | $33.75_{\pm2.46}$ (-22.66) |

which focus on sub-goal decomposition and next-step justification. Moreover, the tokens required for reasoning are reduced to 46.05% on average. This suggests that the test-time reasoning control dynamically adjusts the depth, enabling more efficient inference.

Table 6: Rationale-wise test-time scaling with respect to instruction complexity

| Instruction Template | Plan Length | $z_1$ | $z_2$ | $z_3$ | $z_4$ | $z_5$ | Tokens Reduced |
|---|---|---|---|---|---|---|---|
| TurnOn | 3.80 | 49.4% | 4.5% | 14.8% | 7.2% | 24.1% | 51.1% |
| Open | 3.78 | 48.8% | 7.0% | 5.3% | 12.4% | 26.5% | 48.7% |
| PlaceOn | 6.63 | 35.7% | 11.6% | 9.8% | 18.6% | 24.3% | 44.4% |
| PutIn | 6.95 | 38.1% | 7.8% | 3.5% | 12.7% | 37.8% | 40.0% |

Additionally, in Figure 15 (see Appendix D), we evaluate the effect of *rationale-wise test-time scaling* by applying the same scaling mechanism used in BiCL to SeqFT-Distill. As shown, performance rather decreases for the baseline, as it is trained to predict actions based on the entire rationale set. In contrast, BiCL employs the loss in equation 6, which is optimized to predict actions conditioned on incrementally generated rationales, enabling effective decision-making even from partial rationales.

**Does selecting the most similar prior reasoning-policy yield better performance?** In Table 7, we evaluate two variants of BiCL that use alternative base reasoning-policy selection strategies under the 5-shot setting. *Argmin* selects the policy with the lowest rationale similarity, while *Random* simply uses the random one. These variants do not employ test-time scaling (i.e., TTS in equation 7), thereby isolating the effect of base reasoning-policy selection on the quality of generated rationales. As shown, BiCL w/o TTS which selects the most similar one yields the highest performance, achieving an average SR increase of 5.67% in the seen category and 5.90% in the unseen compared to *Argmin*. Since the current stage's reasoning policy is initialized from the chosen base reasoning-policy, selecting the most similar predecessor provides a strong prior that facilitates efficient adaptation.

Table 7: Effect of base policy selection strategy of BiCL

| Method | VirtualHome Beh-IL | | | | ALFWorld Env-IL | | | |
|---|---|---|---|---|---|---|---|---|
| | Seen | | Unseen | | Seen | | Unseen | |
| | SR (%) | GC (%) | SR (%) | GC (%) | SR (%) | GC (%) | SR (%) | GC (%) |
| *Argmin* | $60.23_{\pm2.09}$ | $74.74_{\pm1.54}$ | $49.12_{\pm2.17}$ | $66.61_{\pm1.82}$ | $51.07_{\pm1.95}$ | $63.59_{\pm1.61}$ | $43.30_{\pm2.47}$ | $56.18_{\pm2.14}$ |
| *Random* | $62.35_{\pm2.04}$ | $74.50_{\pm1.56}$ | $50.79_{\pm2.19}$ | $66.08_{\pm1.86}$ | $54.64_{\pm1.94}$ | $66.29_{\pm1.65}$ | $42.64_{\pm2.45}$ | $57.51_{\pm2.10}$ |
| **BiCL w/o TTS** | $\mathbf{66.33}_{\pm2.14}$ | $\mathbf{77.77}_{\pm1.57}$ | $\mathbf{52.27}_{\pm2.25}$ | $\mathbf{66.33}_{\pm1.92}$ | $\mathbf{57.86}_{\pm2.01}$ | $\mathbf{68.42}_{\pm1.67}$ | $\mathbf{48.21}_{\pm2.58}$ | $\mathbf{60.10}_{\pm2.16}$ |

## 6 Conclusion and Limitations

We presented the BiCL framework tailored for efficient task adaptation of LM-based embodied agents, particularly in scenarios with online access only to limited demonstrations and small LMs. Specifically, our *bidirectional CoT learning* strategy enables effective forward knowledge transfer across adaptation stages, by jointly optimizing CoT reasoning and reflexive reasoning objectives. Notably, it facilitates robust CoT reasoning at inference without requiring explicit reflection steps. Furthermore, the *rationale-wise test-time scaling* mechanism focuses on sufficiently confident rationales, thereby enabling not only more efficient planning but also improved overall performance.

**Limitations.** Our focus is primarily on forward knowledge transfer, leveraging prior reasoning to enhance current learning in a sequential task adaptation setting. Accordingly, BiCL does not explicitly consider backward knowledge transfer, where knowledge gained from new tasks could refine earlier

policies or rationales. Such backward transfer can be addressed by leveraging rehearsal methods (see Appendix D.6), which retain demonstrations across adaptation stages to update the prior policies.

## ETHICS STATEMENT

LM-powered Embodied agents may have broader societal impacts, including unintended behaviors or the amplification of societal biases. Our framework also relies on the reasoning capabilities of LMs, which may introduce incorrect or fabricated information. To mitigate such risks, we maintain full transparency of our rationale annotation pipeline and experimental setup, enabling independent verification and replication. We also emphasize the importance of implementing appropriate safety measures when extending our work beyond simulation and into real-world systems.

## REPRODUCIBILITY STATEMENT

We provide the full benchmark environment settings in Appendix B and the detailed rationale annotation pipeline in Appendix C.1. The complete pseudo-code for our bidirectional CoT learning and rationale-wise test-time scaling is presented in Algorithm 3, and all training hyperparameters are documented in Table 13. We also include the full source code in the supplementary materials. To ensure transparency and reproducibility, we supply comprehensive implementation details, dataset configurations, and benchmark settings, enabling faithful replication of our results.

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

## A    RELATED WORK

**Embodied task planning.** Recently, embodied task planning has gained significant attention, driven by the advancements in LMs' reasoning capabilities (Jiang et al., 2022; Driess et al., 2023; Huang et al., 2022a; 2023). These advancements empower agents to handle complex scenarios such as everyday household tasks, enabling them to produce appropriate plans to accomplish given tasks. SayCan (Ahn et al., 2022) is one of the pioneering works leveraging LMs for embodied tasks, exploiting skill affordance functions to guide the generation of feasible actions. Meanwhile, several works explore the integration of various sources of feedback to either replan or refine decision-making (Huang et al., 2022b; Shinn et al., 2023; Oh et al., 2023). DeDer (Choi et al., 2024) distills the reasoning capabilities of LLMs into small LMs through a two-tier hierarchy, tailored for embodied agents operating on resource-constrained devices. In parallel, a complementary line of research investigates reinforcement learning (RL) as a means of adapting embodied agents across tasks, where agents learn from reward signals through online interaction with the environment (Feng et al., 2025; Chen et al., 2022; Ye et al., 2024). While RL-based approaches demonstrate strong adaptability in simulation, they typically require substantial interactions with the environments, making them difficult to deploy in real-world embodied systems. Our work follows a dataset-driven paradigm, while focusing specifically on enabling LM-based agents to efficiently adapt to new tasks under limited supervision and model capacity.

**Continual task adaptation.** In the domain of continual task adaptation, prior works (Gao et al., 2021; Wan et al., 2024) have focused on leveraging a stream of datasets to progressively learn diverse tasks over time. TAIL (Liu et al., 2024) and L2M (Schmied et al., 2023) adopt parameter-efficient tuning methods to harness the knowledge embedded in pre-trained models for efficient adaptation in robotic manipulation tasks. CAMA (Kim et al., 2024) proposes a framework for continual embodied planning, where model updates are guided by previously stored logits to prevent catastrophic forgetting. Our work distinguishes itself by structuring robust CoT reasoning within LM-based policies, particularly in resource-constrained settings.

**Reasoning distillation and self-correction of LMs.** With the growing reasoning capabilities of LLMs, recent efforts have explored distilling these abilities into smaller models. A common strategy involves extracting CoT rationales from LLMs and using them as supervision signals to train smaller LMs (Li et al., 2023; Wang et al., 2023; Li et al., 2024b; Shridhar et al., 2023; DeepSeek-AI et al., 2025). Furthermore, self-correction mechanisms have emerged as a promising approach to mitigate flawed reasoning in LMs (Saunders et al., 2022). These typically leverage previous responses to bootstrap feedback for self-improvement (Madaan et al., 2023; Sun et al., 2024) or incorporate external feedback from additional knowledge sources (Gou et al., 2024; Shinn et al., 2023). Self-Correction (Welleck et al., 2023) and Aligner (Ji et al., 2024) decouple the initial response generator from a separate corrector, which is trained to refine outputs based on feedback. Unlike previous works that treat CoT reasoning and self-correction as separate capabilities, our BiCL framework aims to enhance CoT reasoning itself through reflexive reasoning. In BiCL, reflexive reasoning internalizes task-specific knowledge by identifying and correcting prior rationales, thus providing richer supervision and enhancing CoT reasoning capabilities for the current task.

## B    BENCHMARK ENVIRONMENTS

### B.1    VIRTUALHOME

VirtualHome (Puig et al., 2018) is a Unity-based simulation environment where an agent interacts with household objects to accomplish given instructions. There are 50 different house settings in VirtualHome, each with different room layouts and object positions. The environment features a wide variety of real-life objects, making it challenging for embodied agents.

**Instruction.** We configure four instruction templates, each with a unique goal to accomplish: TURNON, OPEN, PLACEON, PUTIN. In Table 8, we provide an example for each instruction template.

**Observation.** We use a Sentence-BERT (Reimers & Gurevych, 2019) to retrieve the top-$k$ most relevant triples from the environment knowledge graph based on the given instruction. This setup

Table 8: An example for each instruction template in VirtualHome

| Instruction template | Example |
|---|---|
| TURNON | Turn on tv |
| OPEN | Open cabinet |
| PLACEON | Place apple on sofa |
| PUTIN | Put mug in microwave |

follows similar practices in recent works in embodied agents (Choi et al., 2024; Yoo et al., 2024). In Figure 2, we provide an example of observation.

(character, inside, kitchen), (character, hold, none), (kitchen, adjacent, bedroom), (kitchen, adjacent, livingroom), (bathroom, adjacent, bedroom), (tv, inside, bedroom), (hanger, inside, bedroom), (powersocket, inside, kitchen), (wineglass, inside, kitchen), (kitchencabinet, inside, kitchen), (salmon, inside, kitchen) ...

Figure 2: Observation example for "Turn on tv" task

**Action.** The available actions include: *walk*, *open*, *switch*, *grab*, *place on*, and *put in*. The first four actions take a single argument (e.g. *walk kitchen*, *grab apple*), while the last two actions require two arguments (e.g. *place apple on sofa*, *put mug in microwave*). In Table 9, we present the format of each action along with a corresponding example.

Table 9: Format and example of each action in VirtualHome

| Action | Format | Example |
|---|---|---|
| *walk* | *walk* [object or room] | *walk* kitchen |
| *open* | *open* [object] | *open* cabinet |
| *switch* | *switch* [object] | *switch* tv |
| *grab* | *grab* [object] | *grab* apple |
| *place on* | *place* [object] *on* [object] | *place* apple *on* sofa |
| *put in* | *put* [object] *in* [object] | *put* mug *in* microwave |

**Continual task adaptation.** We configure two continual adaptation setups similar to CAMA (Kim et al., 2024). For Behavior Incremental Learning (Beh-IL), the agent incrementally learns new behaviors; for Environment Incremental Learning (Env-IL), the agent incrementally learns to perform behaviors in novel indoor scenes. In Figure 3, we illustrate these two continual adaptation setups.

For **Beh-IL**, we utilize the four instruction templates described above, assigning each as the primary behavior for a different learning stage. Experiments are conducted across four randomly ordered sequences of tasks, and evaluated under **Seen** and **Unseen** categories. For the Seen category, we use the same tasks and the same room layouts as those used for training; while for the Unseen category, we introduce variations in room layouts, instructions, and initial object positions. The specific sequences used for Beh-IL in VirtualHome are detailed in Figure 4.

For **Env-IL**, we utilize the diverse room layouts provided by VirtualHome, each featuring different object and furniture arrangements. Experiments are conducted across four randomly ordered sequences of these room layouts, and the Seen and Unseen categories are constructed in the same manner as in Beh-IL. The room sequences used for Env-IL in VirtualHome are detailed in Figure 5.

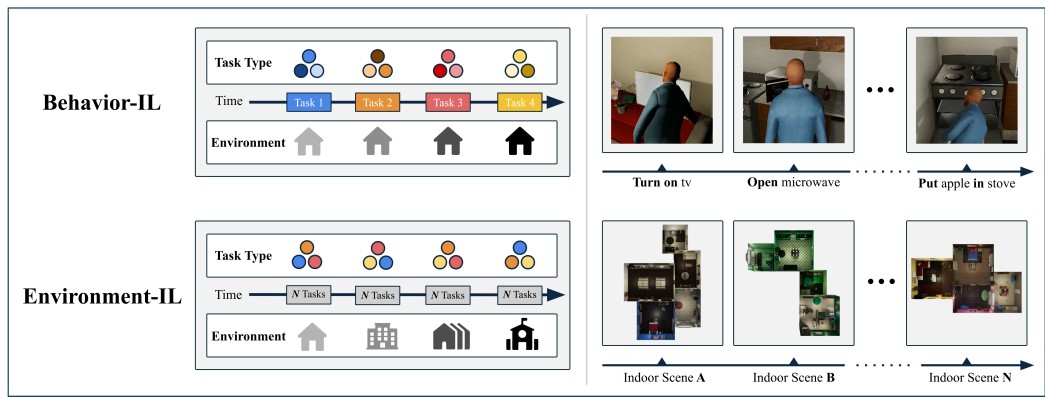

Figure 3: Continual adaptation setups: The top line illustrates Beh-IL, where the agent is tasked to incrementally learn new behaviors. The bottom line illustrates Env-IL, where the agent is tasked to perform behaviors in novel indoor scenes.

> 1. TURNON → OPEN → PLACEON → PUTIN
> 2. OPEN → TURNON → PUTIN → PLACEON
> 3. PLACEON → OPEN → TURNON → PUTIN
> 4. PUTIN → OPEN → PLACEON → TURNON

Figure 4: Continual adaptation sequence for Beh-IL in VirtualHome

> 1. ROOM 20 → ROOM 01 → ROOM 34 → ROOM 26
> 2. ROOM 01 → ROOM 20 → ROOM 26 → ROOM 34
> 3. ROOM 34 → ROOM 01 → ROOM 20 → ROOM 26
> 4. ROOM 26 → ROOM 01 → ROOM 34 → ROOM 20

Figure 5: Continual adaptation sequence for Env-IL in VirtualHome

For the continual few-shot adaptation setting, we double the length of each sequence.

**Expert demonstrations.** To construct few-shot demonstrations, we emulate a rule-based expert that generates an optimal plan based on the given instruction. For each adaptation stage, we collect 20 expert demonstrations, varying the initial positions of the agent and objects, as well as the instructions. For the more constrained continual few-shot adaptation settings, we use 5 expert demonstrations.

### B.2 ALFWORLD

ALFWorld (Shridhar et al., 2021) is a text-based environment designed for embodied task planning, featuring numerous household tasks brought from ALFRED (Shridhar et al., 2020) benchmark. It supports complex task compositions and is widely used to evaluate instruction-following agents under partially observable settings.

**Instruction.** We configure four instruction templates, each with a unique goal to accomplish: HEAT, CLEAN, PICK2&PLACE, PICK&PLACE. In Table 10, we provide an example for each instruction template.

**Observation.** The observation settings are identical to VirtualHome in Section B.1. In Figure 6, we provide an example of observation.

**Action.** The available actions include: *go to*, *take from*, *put in/on*, *heat with*, and *clean with*. The *go to* action takes a single argument specifying the destination (e.g. *go to shelf 1*), while the other

Table 10: An example for each instruction template in ALFWorld

| Instruction template | Example |
|---|---|
| HEAT | Heat some plate in microwave and put it in/on shelf |
| CLEAN | Clean some fork in sinkbasin and put it in/on sidetable |
| PICK2&PLACE | Find two newspaper and put them in/on armchair |
| PICK&PLACE | Put a keychain in/on armchair |

(character, inside, bedroom), (character, hold, keychain 1)
(character, close, armchair 1), (laptop 1, inside, armchair 1),
(lightswitch 1, is, visible), (cellphone 3, inside, bed 1), (cd 2,
inside, garbagecan 1), (pillow 1, inside, bed 1), (cd 1, inside,
dresser 1), (alarmclock 2, inside, dresser 1), (cabinet 1, is, visible) ...

Figure 6: Observation example for task "Put a keychain in/on armchair"

actions require two arguments (e.g. *take potato 1 from countertop 3*, *clean spoon 1 with sinkbasin 1*). In Table 11, we provide the format of each action along with a corresponding example.

Table 11: Format and example of each action in ALFWorld

| Action | Format | Example |
|---|---|---|
| *go to* | *go to* [object] | *go to* shelf 1 |
| *take from* | *take* [object] *from* [object] | *take* potato 1 *from* countertop 3 |
| *put in/on* | *put* [object] *in/on* [object] | *put* newspaper 1 *in/on* ottoman 1 |
| *heat with* | *heat* [object] *with* [object] | *heat* potato 3 *with* microwave 1 |
| *clean with* | *clean* [object] *with* [object] | *clean* spoon 1 *with* sinkbasin 1 |

**Continual task adaptation setup.** As in VirtualHome, we configure two continual task adaptation setups in ALFWorld, similar to CAMA (Kim et al., 2024).

For **Beh-IL**, we utilize the four instruction templates, assigning each as the primary behavior for an adaptation stage. The specific sequences used for Beh-IL in ALFWorld are detailed in Figure 7.

1. HEAT → PICK2&PLACE → PICK&PLACE → CLEAN
2. PICK2&PLACE → CLEAN → HEAT → PICK&PLACE
3. PICK&PLACE → PICK2&PLACE → CLEAN → HEAT
4. CLEAN → HEAT → PICK2&PLACE → PICK&PLACE

Figure 7: Continual adaptation sequence for Beh-IL in ALFWorld

For **Env-IL**, we utilize different room types in ALFWorld, including bathroom, bedroom, livingroom, and kitchen. The specific sequences used for Env-IL in ALFWorld are detailed in Figure 8.

**Expert demonstrations.** To construct the training demonstrations, we utilize the expert provided in ALFWorld. For each adaptation stage, we utilize 20 expert demonstrations, varying the initial positions of the agent and objects, as well as the instructions. For the more constrained continual few-shot adaptation setting, we use 5 expert demonstrations.

1. BEDROOM → BATHROOM → LIVINGROOM → KITCHEN
2. BATHROOM → BEDROOM → KITCHEN → LIVINGROOM
3. LIVINGROOM → BATHROOM → BEDROOM → KITCHEN
4. KITCHEN → BATHROOM → LIVINGROOM → BEDROOM

Figure 8: Continual adaptation sequence for Env-IL in ALFWorld

## C  IMPLEMENTATION DETAILS

In this section, we provide the rationale annotation strategy and implementation details of BiCL and the baselines. All experiments are conducted on a system equipped with an Intel(R) Core(TM) i9-10980XE CPU and an NVIDIA RTX A6000 GPU. We use GPT-4o-mini (Achiam et al., 2023) for rationale annotation and Qwen2.5-0.5B (Yang et al., 2024) as the pre-trained LM of the policies.

### C.1  RATIONALE ANNOTATION

In our implementation, rationales $\mathcal{Z} = \{z_1, ..., z_N\}$ are annotated through LLMs, such as GPT-4o-mini (Achiam et al., 2023) for each observation and action pair $(o, a)$. For this, we define a set of Markov Decision Process-featured queries $\mathcal{Q} = \{q_1, ..., q_N\}$, specifically designed to extract key elements necessary for embodied task planning, such as the location of the target object, the status of the agent, sub-goals, available actions, and expected returns (Choi et al., 2024). We generate 5 multi-step ($N = 5$) rationales, and Table 12 summarizes the key element extracted by each query.

Table 12: Key elements extracted by each query

| Query | Key elements |
|---|---|
| $q_1$ | physical location and status of the agent |
| $q_2$ | physical location and status of observations relevant to the task |
| $q_3$ | summarization of previous action history, if available |
| $q_4$ | sub-goals needed to complete the task |
| $q_5$ | reasoning about the next action to take |

Specifically, we prompt the LLM $\Psi_{\text{LLM}}$ with queries $\mathcal{Q}$, along with the task $\mathcal{T}$, observation $o$, action history $h(= a_{1:t})$, and current action $a(= a_t)$ to sample the rationale set, i.e., $\mathcal{Z} \sim \Psi_{\text{LLM}}(\mathcal{Q}, x, a)$, where $x = (\mathcal{T}, o, h)$. Next, we prompt the policy $\pi(\cdot|\theta_{\text{LM}})$ (i.e., the pre-trained LM without any attached adapters) with the observation and rationale set to obtain logits for the available actions $\bar{a} \in \mathcal{A}$. If the logit corresponding to the ground-truth action $a$ ranks within the top-$k$ action logits, the rationale set $\mathcal{Z}$ is stored in a buffer $\mathcal{B}$. This process is repeated $I$ times with varying temperature and nucleus sampling parameters to generate multiple rationale sets.

$$\mathcal{B} = \bigcup_{i=1}^{I} \{\mathcal{Z} | a \in \text{top-}k_{\bar{a} \in \mathcal{A}} \left( \pi(\bar{a}|o, \mathcal{Z}; \theta_{\text{LM}}) \right) \}, \tag{8}$$
$$\text{where } \mathcal{Z} \sim \Psi_{\text{LLM}}(\mathcal{Q}, x, a).$$

Subsequently, we prompt each rationale set stored in the buffer to the policy to obtain logits for the ground-truth future plan $p(= a_{t:T})$. We then select the rationale set that yields the highest logit as the final rationale set for reasoning-policy training.

$$\mathcal{Z} = \arg\max_{\mathcal{Z}^i \in \mathcal{B}} \pi(p|\mathcal{T}, o, \mathcal{Z}^i; \theta_{\text{LM}}). \tag{9}$$

This entire procedure is performed for each transition in the demonstrations $\mathcal{D}$ to annotate rationales for each sample. Algorithm 2 lists the procedure of rationale annotation. Moreover, Figure 9 presents an example of rationales annotated by the LLM for VirtualHome.

---

**Algorithm 2** Rationale annotation

---

1: **Input:** demonstrations $\mathcal{D}$, LLM $\Psi_{\text{LLM}}$, pre-trained LM parameters $\theta_{\text{LM}}$
2: **for** each $(\mathcal{T}, o, a) \in \mathcal{D}$ **do**
3:      Initialize buffer $\mathcal{B} = \emptyset$
4:      **for** $i \leftarrow 1, ..., I$ **do**
5:         $\mathcal{Z} \sim \Psi_{\text{LLM}}(\mathcal{Q}, x, a)$
6:         **if** $a \in \text{top-}k_{\bar{a} \in \mathcal{A}}(\pi(\bar{a}|o, \mathcal{Z}; \theta_{\text{LM}}))$ **then**
7:            $\mathcal{B} \leftarrow \mathcal{B} \cup \{\mathcal{Z}\}$
8:      $\mathcal{Z} = \arg\max_{\mathcal{Z}^i \in \mathcal{B}} \pi(p|\mathcal{T}, o, \mathcal{Z}^i; \theta_{\text{LM}})$
9:      Augment demonstrations $\mathcal{D}$ with $(\mathcal{T}, o, a, \mathcal{Z})$

---

| | |
|---|---|
| **[Instruction]** | Place book on coffeetable |
| **[Observation]** | (character, inside, kitchen), (character, hold, book), (kitchen, adjacent, bedroom), (kitchen, adjacent, livingroom), (coffeetable, inside, livingroom), (book, is, closed), (character, close, paper), (pie, inside, kitchen), (facecream, inside, bathroom), (closetdrawer, inside, livingroom) ... |
| **[History]** | (step 1, walk bedroom), (step 2, walk kitchen), (step 3, walk book), (step 4, grab book) |
| **[Action]** | walk livingroom |
| **[Rationales]** | The character is currently inside the kitchen, holding a closed book. The coffeetable is located inside the livingroom, which is adjacent to the kitchen. Previously, the character walked from the bedroom to the kitchen, grabbed the book, and is now preparing to move it. To complete the instruction of placing the book on the coffeetable, the character needs to walk to the livingroom next. Since the livingroom is adjacent to the kitchen, the character can proceed there to fulfill the instruction. |

Figure 9: Example of rationales for VirtualHome

## C.2 LLM-PLANNER

LLM-Planner (Huang et al., 2023) leverages the in-context learning capability of language models by prompting few-shot demonstrations. Specifically, two demonstrations are sampled based on the task similarity, which is computed by Jaccard distance. For inference, LLM-Planner employs an action-level decoding strategy to select actions from the set of valid ones (Hazra et al., 2024). Figure 10 presents the prompt template used for LLM-Planner.

## C.3 REACT

ReAct (Yao et al., 2023) generates rationales and actions in an interleaved manner. To allow the LM to better leverage its reasoning capabilities, we prompt it with few-shot in-context CoT demonstrations. In Figure 11, we show the prompt template used for ReAct.

## C.4 SAYCAN

SayCan (Ahn et al., 2022) incorporates an additional module that accounts for affordance scores when selecting actions. This module re-weights the action logits based on predicted affordance scores, ensuring the feasibility of plans and preventing the agent from selecting invalid actions in the current state. In our implementation, we replace this module by heuristically providing only executable actions, filtered based on their feasibility from the current state.

Interact with a household to solve a task. Following are the only actions available:
go to [recep]: move to a receptacle/location
take [obj] from [recep]: pick up an object from a receptacle
put [obj] in/on [recep]: place an object inside or on top of a receptacle
heat [obj] with [recep]: heat an object using a receptacle
clean [obj] with [recep]: clean an object using a receptacle

Here are some examples.

Your task is to: put a pencil in/on desk.
(character, inside, bedroom), (book 3, inside, desk 1), (laptop 1, inside, desk 1), (pencil 2, inside, garbagecan 1), (pencil 1, inside, sidetable 1), (alarmclock 1, inside, desk 1), (remotecontrol 1, inside, desk 1), (alarmclock 2, inside, desk 1), (book 2, inside, desk 1)
> go to sidetable 1
⋮

**[Few-shot Demonstration 2]**
⋮

**[Few-shot Demonstration N]**

Here is the task.

Your task is to: put a pencil in/on desk.
(character, inside, bedroom), (pen 1, inside, desk 1), (laptop 1, inside, desk 1), (pencil 2, inside, sidetable 1), (pencil 3, inside, shelf 1), (keychain 3, inside, desk 1), (box 1, inside, desk 1), (pen 2, inside, desk 1), (book 1, inside, desk 1), (cd 1, inside, garbagecan 1), (cd 2, is, visible), (blinds 1, is, visible), (chair 1, is, visible)
> action:

Figure 10: Prompt example used for LLM-Planner in ALFWorld

## C.5 TAIL

TAIL (Liu et al., 2024) employs parameter-efficient fine-tuning methods to address the continual task adaptation problem. For each adaptation stage, a LoRA adapter is newly initialized and trained on the demonstrations through supervised finetuning. Based on this, we implement two variants for comparison. **TAIL-Action** is trained solely on state-action pairs without utilizing rationale supervision. **TAIL-Distill** leverages rationales to supervise both reasoning and planning policies. The same CoT reasoning loss defined in Equation (3) for BiCL is used to train the reasoning-policy for TAIL-Distill. At inference, rationales are generated segment-by-segment through the reasoning-policy, and subsequently used to generate actions through the planning-policy. The hyperparameter settings for TAIL are summarized in Table 13.

Table 13: Hyperparameter settings for TAIL

| Hyperparameter | Value |
|---|---|
| Total epochs | 80 |
| Batch size | 4 |
| Learning rate | 1.41e-5 |
| LoRA alpha | 32 |
| LoRA rank | 16 |

## C.6 SEQFT-DISTILL

Unlike TAIL, which randomly initializes model parameters for each adaptation stage, SeqFT-Distill uses the policy learned from the most recent adaptation stage as the initialization point. This ensures

---

Interact with a household to solve a task. Following are the only actions available:
walk [object or room]: walk to object or room
open [object]: open an object
switch [object]: switch on an object
grab [object]: grab an object
place [object] on [recep]: put holding on top of an object
put [object] in [recep]: put holding inside of an object

Here are some examples.

Your task is to: Turn on tv.
(character, inside, bedroom), (character, hold, none), (bedroom, adjacent, bathroom), (bedroom, adjacent, kitchen), (kitchen, adjacent, livingroom), (tv, inside, livingroom), (milkshake, inside, livingroom), (curtains, inside, bathroom), (barsoap, inside, bathroom), (fryingpan, inside, kitchen), (kitchencounterdrawer, inside, kitchen)
> think: The character is currently inside the bedroom, which is adjacent to both the bathroom and the kitchen. The TV is located in the living room, which is adjacent to the kitchen but not directly accessible from the bedroom. There have been no prior actions taken that would influence the current state. To turn on the TV, the character needs to walk to the kitchen first, and then from there, proceed to the living room. Given the adjacency of the kitchen to the living room, the next step involves moving towards the kitchen to ultimately reach the TV.
OK.
> walk kitchen
⋮

**[Few-shot Demonstration 2]**
⋮

**[Few-shot Demonstration N]**

Here is the task.

Your task is to: Turn on tv.
(character, close, tv), (character, inside, kitchen), (character, hold, none), (kitchen, adjacent, bedroom), (kitchen, adjacent, livingroom), (bathroom, adjacent, bedroom), (tv, inside, kitchen), (clothesshirt, inside, bedroom), (clothespants, inside, bedroom), (clothesshirt, inside, livingroom), (powersocket, inside, livingroom)
> rationale:
> action:

---

Figure 11: Prompt example used for ReAct in VirtualHome

implicit knowledge transfer across stages through model parameters. For reasoning distillation, we employ both reasoning and planning policies as BiCL. We use the same hyperparameter settings as in Table 13 for SeqFT-Distill.

## C.7 CAMA-DISTILL

CAMA-Distill (Kim et al., 2024) addresses the continual learning problem in embodied tasks by introducing a method to update stored past logits in episodic memory. To accommodate reasoning distillation, we adopt the same two-tier policy architecture as BiCL. A small subset of previous demonstrations, along with their corresponding logits, is retained across adaptation stages and used to supervise the policy alongside the current stage's demonstrations. We use the same hyperparameter settings as in Table 13 for CAMA-Distill.

## C.8 SELF-CORRECTION

Self-Correction Welleck et al. (2023) explicitly learns to iteratively refine imperfect generations. To adapt this approach to our setting, we employ sLMs with adapters for both the initial rationale

---

**Algorithm 3** BiCL framework

---

1: // Adaptation: bidirectional CoT learning
2: **Input:** demonstrations for $i$-th stage $\mathcal{D}_i$, adapter pool $\Theta$
3: Select adapter from adapter pool for base reasoning-policy $\pi_z(\cdot; \theta'_z)$ using (2)
4: Initialize adapters for the current stage $\theta_z \leftarrow \theta'_z, \theta_p$
5: **while** not converged **do**
6:      Sample a batch of $\{(\mathcal{T}, o, a, \mathcal{Z})\} \sim \mathcal{D}_i$
7:      Generate base rationales $z'_k \sim \pi_z(\cdot | x, q_k; \theta'_z)$ through the base reasoning-policy
8:      Update reasoning-policy $\pi_z(\cdot; \theta_z)$ using loss $\mathcal{L}_{\text{reasoning}}$ in (5)
9:      Update planning-policy $\pi_p(\cdot; \theta_p)$ using loss $\mathcal{L}_{\text{planning}}$ in (6)
10: Add adapters to adapter pool $\Theta \leftarrow \Theta \cup \{\theta_z\}$
11: Compute threshold $\delta_k$ for each $k$ using equation 10
12:
13: // Inference: rationale-wise test-time scaling
14: **Input:** environment Env, task $\mathcal{T}$
15: $o_0 \leftarrow$ Env.reset()
16: $done \leftarrow false,\ t \leftarrow 0,\ h \leftarrow \emptyset$
17: **while** not done **do**
18:      $x_t \leftarrow (\mathcal{T}, o_t, h)$
19:      **for** $k \leftarrow 1, ..., N$ **do**
20:          $z_{t,k} \sim \pi_z(\cdot | x_t, q_k; \theta_z)$
21:          $a_t \sim \pi_p(\cdot | x_t, z_{t,1:k}; \theta_p)$
22:          **if** $\log \pi_p(a_t | x_t, z_{t,1:k}; \theta_p) / |a_t| \geq \delta_k$ **then**
23:             break
24:      $o_{t+1} \leftarrow$ Env.step($a_t$)
25:      $h \leftarrow h \cup \{a_t\},\ t \leftarrow t + 1$

---

generator and the corrector. Feedback for correction is obtained in the same way as our setup. The generated rationale is compared with the ground-truth rationales in the demonstrations using a combined similarity score based on language embeddings and TF-IDF scores. At inference, the model first produces an initial rationale and then performs explicit self-correction. At inference, the surrogate feedback is provided by retrieving the rationale in the demonstrations that is most semantically similar to the current state.

## C.9 BiCL (OURS)

The BiCL framework consists of two main processes: (i) bidirectional CoT learning, and (ii) rationale-wise test-time scaling. (i) For adaptation, we first select the most relevant previously learned reasoning-policy to serve as the base one. Then, an LM-based policy is jointly trained via CoT and reflexive reasoning objectives from few-shot demonstrations, where CoT reasoning is supervised by rationale distillation and reflexive reasoning by base rationale correction. We use the same hyperparameter settings as in Table 13 for BiCL. (ii) For inference, the policy solely relies on CoT reasoning, with its depth dynamically adjusted according to the model's confidence in predicted actions.

The threshold $\delta_k$ for rationale-wise test-time scaling in equation 7 is derived from the mean and standard deviation of the log-probability of the ground-truth action when the planning-policy is conditioned on partial rationales $z_{1:k}$ as

$$\delta_k = \mathbb{E}_{(x,a,\mathcal{Z}) \sim \mathcal{D}} \left[ \log \pi_p(a | x, z_{1:k}; \theta_p) / |a| \right] + \lambda \text{std}_{(x,a,\mathcal{Z}) \sim \mathcal{D}} \left[ \log \pi_p(a | x, z_{1:k}; \theta_p) / |a| \right] \quad (10)$$

where $\lambda$ is a hyperparameter corresponding to a desired percentile cutoff, converted into z-score. In our experiment, we set $-0.524$ (corresponds to 30th percentile) for VirtualHome and $0.0$ (corresponds to 50th percentile) for ALFWorld. Thresholds are computed separately at each adaptation stage, yielding a distinct $\delta_k$ for every reasoning step. A sensitivity analysis of this threshold is provided in Section D.7.

The entire procedure for BiCL is summarized in Algorithm 3, and examples of the planning, correction and reasoning prompts are illustrated in Figures 12, 13, and 14, respectively.

### Human:
Following are the only actions available:
walk [object or room]: walk to object or room
open [object]: open an object
switch [object]: switch on an object
grab [object]: grab an object
place [object] on [recep]: put holding on top of an object
put [object] in [recep]: put holding inside of an object

In order to complete the given instruction, what should be the next immediate action?
Instruction: Open stove
State: (character, inside, kitchen), (character, hold, none), (kitchen, adjacent, bedroom), (kitchen, adjacent, livingroom), (bathroom, adjacent, bedroom), (stove, is, closed), (stove, inside, kitchen), (stove, is, off), (toaster, is, off), (closetdrawer, inside, bedroom)
Previous Actions: No action history.
Rationale: The character is currently inside the kitchen, situated near the closed stove. The stove, which needs to be opened, is located inside the kitchen and adjacent to a bedroom. There is no previous action history to consider. To complete the instruction, the character must first walk to the stove. Thus, the logical next action should be walk to stove.

### Assistant:

Figure 12: Planning prompt example used for BiCL in VirtualHome

### Human:
Instruction: heat some cup in microwave and put it in/on sidetable
State: (character, inside, kitchen), (microwave 1, is, visible), (cup 2, inside, countertop 1), (cup 1, inside, sinkbasin 1), (peppershaker 1, inside, sidetable 1), (pot 2, inside, stoveburner 4), (bread 1, inside, countertop 1), (soapbottle 1, inside, garbagecan 1), (spoon 1, is, visible)
Previous Actions: No action history.
Reasoning Trace: The character is currently inside the kitchen. The cup 2 is located on countertop 1, while the microwave 1 is also visible in the kitchen. There is no previous action history to summarize. To complete the instruction, the character needs to first go to countertop 1, retrieve cup 2, heat it in the microwave, and then place it on sidetable 1.
Rationale: Heat some cup in microwave and put it in/on sidetable.
There are many errors in the Think. You need a major revision in the Think. You should provide exactly 1 sentence response that only incorporate: *reasoning for what should do next.*

### Assistant:

Figure 13: Correction prompt example used for BiCL in ALFWorld

**[Query 1]**
### Human:
Instruction: Open stove
State: (character, inside, kitchen), (character, hold, none), (kitchen, adjacent, bedroom), (kitchen, adjacent, livingroom), (bathroom, adjacent, bedroom), (stove, is, closed), (stove, inside, kitchen), (stove, is, off), (toaster, is, off), (closetdrawer, inside, bedroom)
Previous Actions: No action history.
You should provide exactly 1 sentence response that only incorporate: *physical location and status of the character*.

### Assistant:

**[Query 2]**
### Human:
Instruction: Open stove
State: (character, inside, kitchen), (character, hold, none), (kitchen, adjacent, bedroom), (kitchen, adjacent, livingroom), (bathroom, adjacent, bedroom), (stove, is, closed), (stove, inside, kitchen), (stove, is, off), (toaster, is, off), (closetdrawer, inside, bedroom)
Previous Actions: No action history.
You should provide exactly 1 sentence response that only incorporate: *physical location and status of observations that are only related to the instruction*.

### Assistant:

**[Query 3]**
### Human:
Instruction: Open stove
State: (character, inside, kitchen), (character, hold, none), (kitchen, adjacent, bedroom), (kitchen, adjacent, livingroom), (bathroom, adjacent, bedroom), (stove, is, closed), (stove, inside, kitchen), (stove, is, off), (toaster, is, off), (closetdrawer, inside, bedroom)
Previous Actions: No action history.
You should provide exactly 1 sentence response that only incorporate: *summarization of previous action histories if previous actions are available*.

### Assistant:

**[Query 4]**
### Human:
Instruction: Open stove
State: (character, inside, kitchen), (character, hold, none), (kitchen, adjacent, bedroom), (kitchen, adjacent, livingroom), (bathroom, adjacent, bedroom), (stove, is, closed), (stove, inside, kitchen), (stove, is, off), (toaster, is, off), (closetdrawer, inside, bedroom)
Previous Actions: No action history.
You should provide exactly 1 sentence response that only incorporate: *break down the remaining plan to complete the instruction if remaining plans are required*.

### Assistant:

**[Query 5]**
### Human:
Instruction: Open stove
State: (character, inside, kitchen), (character, hold, none), (kitchen, adjacent, bedroom), (kitchen, adjacent, livingroom), (bathroom, adjacent, bedroom), (stove, is, closed), (stove, inside, kitchen), (stove, is, off), (toaster, is, off), (closetdrawer, inside, bedroom)
Previous Actions: No action history.
You should provide exactly 1 sentence response that only incorporate: *reasoning for what should do next*.

### Assistant:

Figure 14: Reasoning prompt example used for BiCL in VirtualHome

# D ADDITIONAL EXPERIMENTS

## D.1 EFFECT OF SEGMENT-WISE REASONING

In Figure 16, we compare two variants of BiCL with different levels of granularity in their reasoning processes. *Full* learns to generate and correct the entire rationale in a single inference step, while *Chunk-wise* processes multiple rationales (two or three) at a time. In contrast, BiCL adopts a segment-wise approach, processing one rationale at an inference step, thus enabling the most fine-grained control over the reasoning process. As shown, finer-grained reasoning consistently leads to performance improvements. This is particularly beneficial for smaller LMs, whose limited capacity benefits from step-by-step guidance.

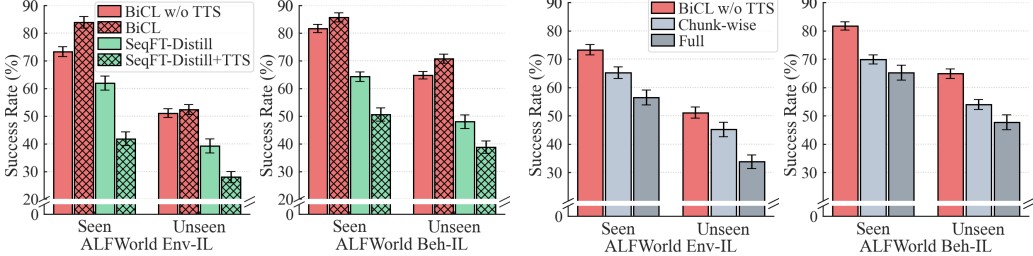

Figure 15: Effect of test-time scaling          Figure 16: Effect of fine-grained reasoning

## D.2 PLANNING EFFICIENCY

To assess the planning efficiency beyond the SR reported in Table 1, we report Normalized Plan Efficiency (NPE) defined as optimal plan length divided by executed plan length (higher is better, and 1.0 indicates an optimal plan) While SR reflects task completion, NPE captures how efficiently the agent completes the given task. A higher NPE indicates more efficient planning, with a value 1.0 representing a perfectly optimal plan.

In Table 14, we report the NPE of BiCL and the baselines under continual task adaptation setups in VirtualHome and ALFWorld. As shown, BiCL consistently achieves higher NPE compared to the baselines, demonstrating its superiority in generating not only successful but also efficient plans. This is attributed to the enhanced robustness of CoT reasoning, enabled by reflexive reasoning that corrects prior knowledge to internalize more precise task knowledge across learning stages.

Table 14: Planning efficiency

| Method | VirtualHome | | | | ALFWorld | | | |
|---|---|---|---|---|---|---|---|---|
| | Beh-IL | | Env-IL | | Beh-IL | | Env-IL | |
| | Seen | Unseen | Seen | Unseen | Seen | Unseen | Seen | Unseen |
| TAIL-Action | $0.79_{\pm 0.02}$ | $0.75_{\pm 0.02}$ | $0.80_{\pm 0.01}$ | $0.77_{\pm 0.01}$ | $0.88_{\pm 0.01}$ | $0.84_{\pm 0.01}$ | $0.87_{\pm 0.01}$ | $0.88_{\pm 0.01}$ |
| TAIL-Distill | $0.80_{\pm 0.01}$ | $0.76_{\pm 0.02}$ | $0.81_{\pm 0.01}$ | $0.73_{\pm 0.01}$ | $0.89_{\pm 0.01}$ | $0.81_{\pm 0.01}$ | $0.91_{\pm 0.01}$ | $0.88_{\pm 0.01}$ |
| SeqFT-Distill | $0.78_{\pm 0.01}$ | $0.72_{\pm 0.02}$ | $0.84_{\pm 0.01}$ | $0.79_{\pm 0.01}$ | $0.91_{\pm 0.01}$ | $0.84_{\pm 0.01}$ | $0.91_{\pm 0.01}$ | $0.89_{\pm 0.01}$ |
| CAMA-Distill | $0.79_{\pm 0.01}$ | $0.78_{\pm 0.01}$ | $0.80_{\pm 0.01}$ | $0.75_{\pm 0.01}$ | $0.89_{\pm 0.01}$ | $0.88_{\pm 0.01}$ | $0.89_{\pm 0.01}$ | $0.86_{\pm 0.01}$ |
| **BiCL w/o TTS** | $0.84_{\pm 0.01}$ | $\mathbf{0.84}_{\pm 0.01}$ | $0.84_{\pm 0.01}$ | $\mathbf{0.81}_{\pm 0.01}$ | $\mathbf{0.95}_{\pm 0.00}$ | $0.93_{\pm 0.01}$ | $0.94_{\pm 0.01}$ | $0.90_{\pm 0.01}$ |
| **BiCL** | $\mathbf{0.85}_{\pm 0.01}$ | $\mathbf{0.84}_{\pm 0.01}$ | $\mathbf{0.87}_{\pm 0.01}$ | $\mathbf{0.81}_{\pm 0.01}$ | $\mathbf{0.95}_{\pm 0.00}$ | $\mathbf{0.94}_{\pm 0.01}$ | $\mathbf{0.95}_{\pm 0.00}$ | $\mathbf{0.92}_{\pm 0.01}$ |

## D.3 CONTINUAL TASK ADAPTABILITY

To further assess continual task adaptability, we evaluate BiCL and the baselines (TAIL-Distill and SeqFT-Distill) on the complete set of tasks from all stages at each individual adaptation stage. Here, we compare with BiCL w/o TTS to isolate the effect of test-time scaling, thereby highlighting the effectiveness of bidirectional CoT learning alone. For tasks from stages not yet encountered, evaluation is performed using the most recently learned policy. As shown, the success rate (SR)

increases roughly linearly for all baselines and BiCL as the number of stages grows, indicating that the agents incrementally acquire new behaviors and adapt to novel scenes. However, the performance gain of BiCL becomes more pronounced with increasing adaptation stages, achieving a 5.50% improvement at stage 2 and a 13.96% improvement at stage 4 on the seen category in VirtualHome Beh-IL compared to SeqFT-Distill. This advantage stems from our bidirectional CoT learning, which enables more effective forward transfer of previously acquired knowledge to new tasks.

Table 15: Continual Task Adaptability measured by SR (%)

| Method | Stage 1 | | Stage 2 | | Stage 3 | | Stage 4 | |
|---|---|---|---|---|---|---|---|---|
| | Seen | Unseen | Seen | Unseen | Seen | Unseen | Seen | Unseen |
| TAIL-Distill | $20.83_{\pm0.92}$ | $15.21_{\pm0.92}$ | $33.55_{\pm0.87}$ | $26.27_{\pm1.08}$ | $49.80_{\pm1.74}$ | $36.38_{\pm1.83}$ | $60.25_{\pm2.20}$ | $46.29_{\pm2.25}$ |
| SeqFT-Distill | $19.13_{\pm0.91}$ | $13.89_{\pm0.87}$ | $32.75_{\pm1.00}$ | $23.81_{\pm1.13}$ | $49.33_{\pm1.67}$ | $37.26_{\pm1.83}$ | $62.17_{\pm2.22}$ | $46.31_{\pm2.22}$ |
| **BiCL w/o TTS** | $\mathbf{24.63}_{\pm0.91}$ | $\mathbf{19.78}_{\pm0.98}$ | $\mathbf{38.25}_{\pm1.01}$ | $\mathbf{29.42}_{\pm1.30}$ | $\mathbf{56.63}_{\pm1.61}$ | $\mathbf{44.79}_{\pm1.96}$ | $\mathbf{76.13}_{\pm1.91}$ | $\mathbf{60.29}_{\pm2.32}$ |

## D.4 EXPERIMENTS WITH LARGER sLMs

To validate that the BiCL framework naturally scales to larger sLMs, we evaluate BiCL and SeqFT-Distill on VirtualHome Beh-IL using **Qwen2.5-1.5B** and **Qwen2.5-3B** in Table 16. Consistent with the results in Table 1, on Qwen2.5-0.5B, BiCL outperforms the strongest baseline SeqFT-Distill in the seen category, achieving SR gains of 14.99% on the 1.5B model and 13.75% on the 3B model. Similarly, in the unseen category, BiCL surpasses SeqFT-Distill with SR improvements of 16.29% on the 1.5B model and 14.43% on the 3B model. These results demonstrate that BiCL scales robustly, yielding consistent performance improvements as model size increases.

Table 16: Performance with larger sLMs (0.5B, 1.5B, 3B)

| Method | Seen | | Unseen | |
|---|---|---|---|---|
| | SR (%) | GC (%) | SR (%) | GC (%) |
| **Model: Qwen2.5-0.5B** | | | | |
| **BiCL** | $\mathbf{81.38}_{\pm1.74}$ | $\mathbf{85.81}_{\pm1.36}$ | $\mathbf{64.03}_{\pm2.28}$ | $\mathbf{71.60}_{\pm3.03}$ |
| SeqFT-Distill | $65.75_{\pm2.15}$ (-15.63) | $79.09_{\pm1.52}$ (-6.72) | $48.68_{\pm2.30}$ (-15.35) | $64.11_{\pm1.92}$ (-7.49) |
| **Model: Qwen2.5-1.5B** | | | | |
| **BiCL** | $\mathbf{85.12}_{\pm1.52}$ | $\mathbf{89.28}_{\pm1.14}$ | $\mathbf{68.64}_{\pm2.12}$ | $\mathbf{75.16}_{\pm1.87}$ |
| SeqFT-Distill | $70.13_{\pm2.01}$ (-14.99) | $79.88_{\pm1.49}$ (-9.40) | $52.34_{\pm2.30}$ (-16.30) | $64.41_{\pm1.96}$ (-10.75) |
| **Model: Qwen2.5-3B** | | | | |
| **BiCL** | $\mathbf{87.00}_{\pm1.37}$ | $\mathbf{91.91}_{\pm0.91}$ | $\mathbf{72.37}_{\pm2.09}$ | $\mathbf{78.07}_{\pm1.80}$ |
| SeqFT-Distill | $73.25_{\pm2.08}$ (-13.75) | $83.28_{\pm1.43}$ (-8.63) | $57.94_{\pm2.34}$ (-14.43) | $69.22_{\pm1.92}$ (-8.85) |

## D.5 EXPERIMENTS WITH COMPOSITIONAL TASKS

To evaluate on more challenging tasks, we design compositional tasks in VirtualHome, where the agent should complete the two or three instructions in sequence, such as "Turn on computer, and turn on radio" (composition of two instructions) or "Put apple in fridge, and turn on stove, and place paper on bed" (composition of three instructions). As shown in Table 17, BiCL outperforms the most competitive baseline SeqFT-Distill, achieving a 7.29% SR gain in the seen category and a 15.65% gain in the unseen category. These results highlight the capability of BiCL to adapt even in complex, compositional tasks.

## D.6 INCORPORATING REHEARSAL STRATEGY FOR BACKWARD TRANSFER

BiCL can be seamlessly extended to support backward transfer through memory-based rehearsal strategies (Rolnick et al., 2018; Wan et al., 2024). To demonstrate this, we introduce a variant of BiCL with a rehearsal mechanism (BiCL w/ Rehearsal), in which demonstrations are retained

Table 17: Performance on Compositional Tasks

| Method | Seen | | Unseen | |
|---|---|---|---|---|
| | SR (%) | GC (%) | SR (%) | GC (%) |
| SayCan | $22.08_{\pm0.98}$ | $48.18_{\pm0.90}$ | $15.84_{\pm0.75}$ | $45.18_{\pm0.80}$ |
| SeqFT-Distill | $48.12_{\pm1.64}$ | $69.85_{\pm1.49}$ | $31.00_{\pm1.15}$ | $57.94_{\pm1.29}$ |
| **BiCL w/o TTS** | $50.59_{\pm1.67}$ | $73.68_{\pm1.63}$ | $40.56_{\pm1.02}$ | $64.12_{\pm1.31}$ |
| **BiCL** | $\mathbf{55.41}_{\pm1.55}$ | $\mathbf{77.13}_{\pm1.56}$ | $\mathbf{46.65}_{\pm0.92}$ | $\mathbf{71.28}_{\pm1.09}$ |

across adaptation stages. After the final stage, BiCL w/ Rehearsal refines earlier policies using the combined CoT reasoning and reflexive reasoning loss defined in equation 5. For each policy, the most semantically relevant policy (excluding itself) is chosen from the learned pool as the base policy, and training is performed on both the original demonstrations for that stage and task-relevant demonstrations accumulated in the rehearsal buffer.

Table 18 reports results under the continual 5-shot adaptation setup in VirtualHome Beh-IL. As shown, incorporating rehearsal mechanism improves performance, yielding SR gains of 5.06% in the seen category and 10.68% in the unseen. These findings demonstrate that BiCL can effectively leverage memory-based rehearsal to enhance backward transfer.

Table 18: Performance of BiCL with rehearsal mechanism

| Method | Seen | | Unseen | |
|---|---|---|---|---|
| | SR (%) | GC (%) | SR (%) | GC (%) |
| **BiCL** | $70.16_{\pm2.04}$ | $78.83_{\pm1.58}$ | $47.99_{\pm2.20}$ | $63.91_{\pm1.96}$ |
| **BiCL w/ Rehearsal** | $\mathbf{75.22}_{\pm1.82}$ **(+5.06)** | $\mathbf{81.33}_{\pm1.48}$ **(+2.50)** | $\mathbf{58.67}_{\pm2.31}$ **(+10.68)** | $\mathbf{68.79}_{\pm1.98}$ **(+4.88)** |

### D.7 SENSITIVITY TO RATIONALE-WISE TEST-TIME SCALING THRESHOLD

To assess the sensitivity to the threshold $\delta_k$ used for rationale-wise test-time scaling in equation 7, we vary the threshold levels in VirtualHome Beh-IL by adjusting $\lambda$ in equation 10. In Table 19, *Low* denotes the default threshold setting used in our main experiments. As shown, applying test-time scaling generally yields higher performance than not applying it. Setting the threshold too low (e.g., *Very Low*) prematurely halts CoT reasoning, resulting in a performance drop of 3.76% in the seen category and 5.84% in the unseen compared to the *low* (default) setting. Setting the threshold too high (e.g. *Very High*) also leads to degradation, with drops of 6.26% in the seen category and 2.49% in the unseen. We conjecture that limited data at each adaptation stage leaves some rationales under-optimized, and forcing the model to "over think" compounds errors and degrades overall performance (Liu et al., 2025). Nevertheless, BiCL consistently outperforms all baselines in Table 1 across all threshold values while maintaining superiority in computational efficiency. Notably, even under the worst-performing threshold conidtion (Very High), BiCL still exceeds the most competitive baseline, SeqFT-Distill, by 9.37% on the seen category and 12.41% on the unseen category.

Table 19: Sensitivity analysis on rationale-wise test-time scaling thresholds

| Threshold Level | Seen SR (%) | Unseen SR (%) |
|---|---|---|
| w/o TTS | $76.13_{\pm1.91}$ | $60.29_{\pm2.32}$ |
| Very Low | $77.62_{\pm1.88}$ | $57.74_{\pm2.29}$ |
| Low (Default) | $\mathbf{81.38}_{\pm1.74}$ | $\mathbf{63.58}_{\pm2.28}$ |
| High | $79.25_{\pm1.82}$ | $61.35_{\pm2.01}$ |
| Very High | $75.12_{\pm1.98}$ | $61.09_{\pm2.29}$ |

