# OpenReview forum: "Asymmetric Effects of Self-Corrective Learning on Chain-of-Thought Reasoning for Efficient Policy Adaptation"
_ICLR.cc/2026/Conference — Submitted to ICLR 2026_

### Official Review · Reviewer_6u9M · 2025-10-29

**Soundness:** 2
**Presentation:** 3
**Contribution:** 2
**Rating:** 4
**Confidence:** 4

**Summary:**

This paper addresses LM continual learning across task variations by proposing a method that jointly optimizes CoT reasoning and reflexive reasoning using few-shot expert demonstrations. To improve inference efficiency, the proposed method is further equipped with rationale-wise test-time scaling to dynamically adjust the depth of CoT reasoning, which is achieved by evaluating model's logits on ground-truth actions conditioned on generatated rationales.

Experiments on VirtualHome and ALFWorld shows the effectiveness of proposed method (with test-time scaling) compared to several continual task adaptation approaches using small LMs, though it still underperforms LLMs (GPT-4o) prompted with demonstrations.

**Strengths:**

1. Simultenously fine-tuning CoT and reflexive reasoning in LMs instead of treating them as separate capabilities is insighful, and the findings about the asymmetric effects of reflexive reasoning on CoT reasoning is promising: reflecive reasoning can enhance CoT reasoning through LM fine-tuning but it does not require explicit reflection during inference.
2. Dynamically adjust the depth of CoT reasoning during inference is crutial to computation efficiency, though the paper relies on ground-truth actions to achieve this.
3. The paper is well-written and extensive experiments have been conducted to demonstrate the effectiveness of proposed method in different aspects, including inference efficiency and training data efficiency.

**Weaknesses:**

1. **Heavily relies on expert demonstrations.**
Each component of BiCL including the rationale-wise TTS relies heavily on expert demonstrations consisting of CoT rationales and ground-truth actions, which is impractical in many realistic embodied tasks, especially those involving open-ended environments without any ground-truth labels.
2. **Concerns about the usefulness of the method in practice.**
In the paper, the rationales in demonstrations are annotated by GPT-4o-mini, which still relies on LLMs despite that the paper assums a resource-constrained setting. Even so, fine-tuning a small LM (qwen-0.5b) via LoRA with the distilled CoT (from GPT-4o-mini) still underperforms GPT-4o, raising concerns about the significance of using such a method in practice.
3. **Assumed resource-constrained settings does not aligned with the experiments.**
Benchmarks used in the experiments, VirtualHome and ALFWorld, does not specifically aim for resource-constrained settings in embodied tasks, so it is not aligned with the paper's assumption that only small LMs (0.5b, 1.5b, 3b) can be available. It is noted that a qwen2.5-7b model trained with RL has achieved nearly perfect performance in ALFWorld [1].
4. **Lack of ablation studies to validate each component of BiCL.**
Though the paper has provided some ablation results, but they are separate, making it hard to find what component contributes most and least to BiCL.
5. Better to provide ethics and reproducibility statement.

References:

[1] Feng, Lang et al. “Group-in-Group Policy Optimization for LLM Agent Training.” NeurIPS 2025.

**Questions:**

1. What does "generate rationales segment by segment" in line 208 mean?
2. Please elaborate how to acquire the feedback $f (z_k, z^′_k)$ in details.
3. Please elaborate the notion of CoT step, action execution step, and the task index as welll as their relationships. For example, within a task $i$, does a trajectory contain several action steps and before each action step, $k$ CoT steps are generated? How to define a CoT step?
4. Please elaborate more on **BiCL w/o base**. If it does not leverage the base reasoning-policy in any form, then what policy does it use?
5. How to calculate the percentage of the rationale $z_k$ at which reasoning is terminated in Table 6?

---

> ### Author Response · Authors · 2025-11-19
>
> **`[Weakness 1] Heavily relies on expert demonstrations`**
>
> In our setup, **BiCL rely on few-shot demonstrations for each adaptation stage (e.g., 20-shot in Table 1, and 5-shot in Table 2)** rather than large-scale datasets. This keeps the supervision cost extremely low and aligns with realistic scenarios where only a small amount of demonstrations can typically be collected. We believe that this setup mitigates the reviewer’s concern regarding impractical data requirements and better reflects the intended data-efficient adaptation scenario studied in our work.
>
> ----
>
> **`[Weakness 2] Concerns about the usefulness of the method in practice`**
>
>  We would like to clarify the meaning of the resource-constrained setting in our paper. **The constraint refers to the compute limitations at deployment time, where embodied agents (e.g., robots or edge devices) must operate with small language models due to limited onboard GPU/CPU capacity.** Accordingly, in our setup, we assume the LLM is used only server-side training to distill reasoning capabilities, while all inference after deployment is handled solely by the sLM.
>
> In such resource-constrained scenarios we target, LLMs (e.g., GPT-4o) cannot be deployed on the agent in the first place. **Thus, the relevant question is not whether a distilled sLM can match the teacher LLM, but whether it can deliver practical performance relative to other sLM-based baselines in settings where LLMs are infeasible to deploy.** In this regard, as shown in Table 1, our experiments demonstrate that BiCL achieves substantial improvements over sLM-based planning baselines (e.g., ReAct, SayCan) and sLM-based continual task adaptation baselines (e.g., TAIL-Distill, CAMA-Distill, SeqFT-Distill). Moreover, even compared to the GPT-4o, BiCL exhibits modest performance gap of 9.57\% at average, despite only using a 0.5B-parameter model and few-shot demonstrations for training. This indicates that BiCL offers a highly practical solution for real-world, compute-constrained embodied agents where deploying LLM is simply infeasible.
>
> ----
>
> **`[Weakness 3] Assumed resource-constrained settings does not aligned with the experiments.`**
>
> As clarified in Weakness 2, **the resource-constrained setting in our paper refers to the computational limitations of deployment platforms (e.g., robots or edge devices), not to the properties of the benchmark environments**. VirtualHome and ALFWorld are used purely as embodied task testbeds for evaluating embodied reasoning and planning capabilities.
>
> **This design principle is standard in recent LM-based planning and VLA systems, which commonly adapt small-sized models for real-world applicability.** For example, PRSIM [1] uses 3B model, DEDER [2] uses 0.8B model, molVLA [3] uses 0.45B model, pi0 [4] uses 3.3B model. Consistent with these works, our default model size is also a small LM (0.5B), aligning with realistic deployment constraints.
>
> Regarding to the statement that ``Qwen2.5-7B trained with RL has achieved nearly perfect performance in ALFWorld'', we emphasize that this result is not direcly comparable due to fundamental differences in supervision scale and learning paradigm. **The cited work [5] requires 19,200 training episodes in ALFWorld; whereas, our framework performs data-efficient adaptation using only 20-episode (Table 1) or even 5-episode (Table 2) demonstrations.** Given this substantial difference in supervision scale, it is expected that an RL-trained 7B model can reach near-perfect performance. **Moreover, the result is not directly comparable, as the cited work employs reinforcement learning, which trains the model through online interaction, whereas our method uses dataset-based supervised fine-tuning.** Therefore, the cited result does not contradict our claims; instead, it highlights that achieving strong performance under strict data and model-size constraints is precisely the challenge BiCL is designed to address.
>
> [1] Distilling On-device Language Models for Robot Planning with Minimal Human Intervention, CoRL 2025
>
> [2] Embodied CoT Distillation From LLM To Off-the-shelf Agents, ICML 2024
>
> [3] SmolVLA: A Vision-Language-Action Model for Affordable and Efficient Robotics, 2025
>
> [4] $\pi_0$: A Vision-Language-Action Flow Model for General Robot Control, 2025
>
> [5] Feng, Lang et al. “Group-in-Group Policy Optimization for LLM Agent Training.” NeurIPS 2025.

---

> ### Author Response · Authors · 2025-11-19
>
> **`[Weakness 4] Lack of ablation studies to validate each component of BiCL.`**
>
> While the ablation studies analyze all individual components, we notice that these analyses are presented across separate tables, which may make it less straightforward to understand their relative contribution. To address this concern, we provide clarifications below.
>
> **Our submission reports the ablation for all key components in BiCL.** (i) The reflexive reasoning loss is ablated in Table 5, showing the largest performance drop of up to 20.54\% in SR when BiCL does not employ the reflexive reasoning loss. (ii) base reasoning-policy selection strategy is ablated in Table 7, demonstrating consistent gains of 5-6\% compared to alternative minimum-similarity or random selection strategies. (iii) the rationale-wise test-time scaling is ablated in the Table 3, exhibiting clear improvements in both performance and computation efficiency when TTS is applied.
>
> ----
>
> **`[Weakness 5] Better to provide ethics and reproducibility statement.`**
>
> **We include the following as the ethics statement in the revised manuscript.**
>
> LM-powered Embodied agents may have broader societal impacts, including unintended behaviors or the amplification of societal biases. Our framework also relies on the reasoning capabilities of LMs, which may introduce incorrect or fabricated information. To mitigate such risks, we maintain full transparency of our rationale annotation pipeline and experimental setup, enabling independent verification and replication. We also emphasize the importance of implementing appropriate safety measures when extending our work beyond simulation and into real-world systems.
>
> **We include the following as the reproducibility statement in the revised manuscript.**
>
> We provide the full benchmark environment settings in Appendix B and the detailed rationale annotation pipeline in Appendix C.1. The complete pseudo-code for our bidirectional CoT learning and rationale-wise test-time scaling is presented in Algorithm 3, and all training hyperparameters are documented in Table 13. We also include the full source code in the supplementary materials. To ensure transparency and reproducibility, we supply comprehensive implementation details, dataset configurations, and benchmark settings, enabling faithful replication of our results.
>
> ----
>
> **`[Question 1]  What does "generate rationales segment by segment" in line 208 mean?`**
>
> "Generate rationales segment by segment" means that **the model produces a multi-step CoT not as a single long rationale, but as a sequence of rationale segments generated one at a time**. In our formulation, each rationale segment $z_k$ corresponds to the response for a predefined query $q_k$. During inference, the model generate $z_1$ in resonse to $q_1$, then $z_2$ in response to $q_2$, and continues this process sequentially for all $k$ steps. Such a segment-by-segment generation mitigates cascading errors across CoT steps, which is particularly important under few-shot supervision and given the limited reasoning capacity of sLMs.
>
> ----
>
> **`[Question 2]  Please elaborate how to acquire the feedback $f(z_k,z'_k)$ in details.`**
>
> To commute the feedback $f(z_k, z'_k)$, we first obtain the sentence embeddings of the generated rationale $z_k$ and the base rationale $z'_k$ using the MiniLM-L6-v2, and calculate their cosine similarity. In addition, we compute a TF-IDF similarity score between the two rationales. The final similarity score is defined as the sum of these two similarity terms. Based on this combined similarity score, we assign the feedback category using threshold-based rules. If the similarity is at least 0.82, the rationale is labeled as a *minor revision*; if it falls between 0.6 and 0.82, it is labeled as a *moderated revision*; and if it is below 0.6, it is categorized as a *major revision*.

---

> ### Author Response · Authors · 2025-11-19
>
> **`[Question 3]  Please elaborate the notion of CoT step, action execution step, and the task index as welll as their relationships`**
>
> Our formulation follows the standard Markov Decision Process (MDP) setting, where **completing a task $i$ requires executing a sequence of action steps within an episode**. At each action step $t$, the agent first performs CoT reasoning before producing the action. In our setup, CoT reasoning consists of generating rationales correspond to a pre-defined $k$ queries. Accordingly, a **CoT step refers to generating a single rationale $z_k$ in response to its corresponding query $q_k$. After completing all $k$ CoT steps and obtaining the full set of rationales, the agent uses them to predict the action for step $t$.**
>
> This reasoning-then-acting cycle is repeated at every action step, which follows common practice in embodied ai literature [1-4]. The queries used for CoT reasoning is provided in Table 12, and the example rationales are shown in Figure 9. For further clarification, we also provide the specific query list and corresponding rationale examples below.
>
> | $k$ | Query | Example Rationale |
> |--|--|--|
> |1| $q_1$ : physical location and status of the agent | $z_1$ : The character is currently inside the kitchen, holding a closed book. |
> |2| $q_2$ : physical location and status of observations relevant to the task | $z_2$ : The coffeetable is located inside the livingroom, which is adjacent to the kitchen.|
> |3| $q_3$ : summarization of previous action history, if available | $z_3$ : Previously, the character walked from the bedroom to the kitchen, grabbed the book, and is now preparing to move it.|
> |4| $q_4$ : sub-goals needed to complete the task | $z_4$ : To complete the instruction of placing the book on the coffeetable, the character needs to walk to the livingroom next. |
> |5| $q_5$ : reasoning about the next action to take | $z_5$ : Since the livingroom is adjacent to the kitchen, the character can proceed there to fulfill the instruction
>
> [1] Shinn, Noah, et al. "Reflexion: Language agents with verbal reinforcement learning." NeurIPS 2023
>
> [2] Zawalski, Michał, et al. "Robotic control via embodied chain-of-thought reasoning." CoRL 2024
>
> [3] Huang, Wenlong, et al. "Inner monologue: Embodied reasoning through planning with language models." CoRL 2022
>
> [4] Yao, Shunyu, et al. "React: Synergizing reasoning and acting in language models."  ICLR 2022
>
> ----
>
> **`[Question 4] Please elaborate more on BiCL w/o base. If it does not leverage the base reasoning-policy in any form, then what policy does it use?`**
>
> In the \textit{w/o base} variant, the reasoning-policy is **neither initialized from the base reasoning-policy nor employs the reflexive reasoning loss**. Instead, the reasoning-policy is initialized from the pretrained LM with a LoRA adapter and is optimized solely through the CoT reasoning objective. Conceptually, this variant is similar to TAIL-Distill, with one key difference. While TAIL-Distill generates the full rationale in a single pass, the w/o base variant follows BiCL's segment-by-segment rationale generation scheme.
>
> ----
>
> **`[Question 5]  How to calculate the percentage of the rationale $z_k$ at which reasoning is terminated in Table 6?`**
>
> To compute the percentage of the rationale $z_k$ at which reasoning is terminated, we run BiCL with rationale-wise test-time scaling and record the index $k$ of the rationale segment where the model first decides to stop further reasoning. Specifically, **when the planning-policy's confidence first exceed the threshold $\delta_k$, we treat this point as ``termination at $z_k$''**. For each instruction template, we then count how many action steps terminate at each $k \in {1, ..,5}$, and divide this count by the total number of action steps for that template.
>
> ---
>
> **`[Summary Response]`**
>
> We sincerely thank the reviewers for their insightful and constructive feedback. We have carefully addressed all comments and provided detailed clarifications, especially regarding the definition of the resource-constrained setting in our work and the implementation details raised in the questions. If there are any additional questions or suggestions, we would be very glad to provide further explanations.

---

> > ### Comment · Reviewer_6u9M · 2025-11-25
> >
> > Thanks for the detailed response, and I would require the authors to update their paper with all these implementation details.
> >
> > My main concern is that this paper situates its method within a very limited scope of existing literature on task adaptation for embodied agents. In particular, RL is a standard approach for enabling LLM-based agents to adapt across tasks without supervised data, yet it is not addressed in either the related work or the baselines. For this reason, I am keeping my original score.

---

> ### Author Response · Authors · 2025-11-25
>
> Reinforcement learning (RL) is indeed one possible approach for training embodied agents. However, our work operates under a offline imitation-learning setting, where the policy must adapt from a fixed offline dataset without any online interaction. To address the reviewer’s suggestion, we have added a discussion of RL-based embodied-agent approaches in Appendix A and Section 2 (Related Works), along with a clarification of how our formulation differs from and complements these methods.
>
> In practice, **deploying RL in real-world emobided systems is extremely challenging due to the need for continuous environment interaction**, carefully designed reward functions, safe exploration, and a very large number of training episodes (e.g., RL-based ALFWorld agents require 19,200 training episodes [1], whereas our imitation-learning approach achieves adaptation with only 20-shot episodes.).
>
> Accordingly, **a growing line of work adopts offline, data-centric learning paradigms similar to ours (e.g., SayCan [2], VIMA [3], Open-X [4], RAEA [5], SayCanpay [6], DEDER [7], LLM-Planner [8])**. These works leverage datasets in various ways, such as learning the policy itself [3,7], training affordance functions [2,6], constructing replay buffers [5], or providing in-context examples for planning [8]. These approaches adopt datasets as a primary source of supervision, explicitly avoiding online RL due to its impracticality in real robotic systems.
>
> Therefore, our formulation reflects a realistic and widely adopted deployment scenario for embodied agents, consistent with recent state-of-the-art embodied systems that rely on offline instead of reward-driven RL.
>
> We once again sincerely appreciate the reviewer’s feedback and the time to evaluating our work. We hope this helps clarify that our formulation reflects practical constraints, and aligns with a substantial and growing body of recent research adopting offline, data-driven approaches. We would be grateful if the reviewer could consider these points in re-evaluating our work.
>
> [1] Feng, Lang et al. “Group-in-Group Policy Optimization for LLM Agent Training.” NeurIPS 2025
>
> [2] Ahn, Michael, et al. "Do as i can, not as i say: Grounding language in robotic affordances."  CoRL 2022
>
> [3] Jiang, Yunfan, et al. "Vima: General robot manipulation with multimodal prompts." ICML 2023
>
> [4] Vuong, Quan, et al. "Open x-embodiment: Robotic learning datasets and rt-x models." CoRL 2023
>
> [5] Zhu, Yichen, et al. "Retrieval-augmented embodied agents." CVPR 2024
>
> [6] Hazra, Rishi, Pedro Zuidberg Dos Martires, and Luc De Raedt. "Saycanpay: Heuristic planning with large language models using learnable domain knowledge." AAAI 2024
>
> [7] Choi, Wonje, et al. "Embodied cot distillation from llm to off-the-shelf agents." ICML 2024
>
> [8] Song, Chan Hee, et al. "Llm-planner: Few-shot grounded planning for embodied agents with large language models." ICCV 2023

---

### Official Review · Reviewer_LD91 · 2025-10-31

**Soundness:** 3
**Presentation:** 3
**Contribution:** 3
**Rating:** 6
**Confidence:** 3

**Summary:**

The paper addresses continual task adaptation for LM-based embodied agents under limited supervision and small model capacity. It proposes BiCL, a framework which jointly optimizes CoT reasoning and reflexive reasoning via few-shot demonstrations and rationale-wise correction of prior policie. Moreover, it introduces rationale-wise test-time scaling to dynamically adjust reasoning depth. Experiments on VirtualHome and ALFWorld show performance gains over baselines in data, computation, and parameter efficiency.

**Strengths:**

- The paper is generally well-written and easy to follow.

- I think the idea of Introducing reflexive reasoning as correction of prior policy rationales, thus enabling forward knowledge transfer without inference-time reflection, is novel and interesting.

- Experimental results on VirtualHome and ALFWorld show consistent performance improvement over previous baselines in data, computation, and parameter efficiency.

**Weaknesses:**

- Token reduction is reported for BiCL, but no wall-clock time or FLOPs are provided.

- Task sequences are fixed to 4 stages; scalability to longer streams is not assessed.

- No ablation is provided to test whether the categorical feedback is necessary for reflection—e.g., whether using binary feedback (correct/incorrect) or the original design (minor/moderate/major revision) is better.

**Questions:**

Please see the weaknesses for the major concerns. I recommend for weak acceptance.

---

> ### Author Response · Authors · 2025-11-22
>
> `Weakness 1) Token reduction is reported for BiCL, but no wall-clock time or FLOPs are provided.`
>
> To further demonstrate the computational efficiency of BiCL, we additionally report the wall-clock time and FLOPs of BiCL and BiCL w/o TTS below. All measurements are conducted on the VirtualHome Beh-IL Benchmar using a system equipped with an Intel(R) Core(TM) i9-109800XE CPU and an NVIDIA RTX A6000 GPU. As shown, **BiCL reduces token generation by 48.2\%, wall-clock time by 49.2\%, and FLOPs by 48.2\% compared to BiCL w/o TTS**, indicating that reductions in token generation directly translates into substantial computational savings. We will include these results in the revised manuscript.
>
> | Method| Generated Tokens | Wall-Clock Time | FLOPs
> |--|--|--|--|
> | BiCL w/o TTS| 84.3 | 4.49 sec  | 5.62T  |
> | BiCL| 40.6 (48.2\% reduced) | 2.21 sec (49.2\% reduced)   | 2.71T (48.2\% reduced) |
>
> ----
>
> `Weakness 2) Task sequences are fixed to 4 stages; scalability to longer streams is not assessed`
>
> We thank the reviewer for pointing this out. While Table 1 reports results under a 4-stage continual adaptation setting, **Table 2 evaluates BiCL under a 7-stage continual adaptation setting**. We inadvertently omitted this clarification in the main text. In the revised manuscript, we have added a sentence to each table's descriptive paragraph to explicitly state the corresponding number of adaptation stages.
>
> ----
>
> `Weakness 3) No ablation is provided to test whether the categorical feedback is necessary for reflection`
>
> In the table below, **we provide additional ablations evaluating whether categorical feedback is necessary for effective reflection.** Specifically, we compare our original three-level feedback design (minor / moderate / major revision) with a binary feedback variant of BiCL that uses only a correct vs. incorrect signal during training, denoted as BiCL *with binary feedback*. As shown, the categorical feedback consistently outperforms the binary variant in VirtualHome Beh-IL, yielding a 2.63% higher SR on the seen category and a 3.73% higher SR on the unseen category. This demonstrates that finer-grained feedback leads to improved performance.
>
> | Method| Seen SR (%) | Unseen SR (%) |
> |--|--|--|
> | BiCL with binary feedback | 78.75 ± 1.71 | 60.30 ± 2.28 |
> | BiCL| 81.38 ± 1.74 | 64.03 ± 2.28 |
>
> ----
>
> **`[Summary Response]`**
>
> We sincerely thank the reviewers for their insightful and constructive feedback. We have carefully addressed all comments and incorporated detailed clarifications. If there are any further questions or suggestions, we would be very glad to provide additional explanations.

---

> > ### Comment · Reviewer_LD91 · 2025-11-22
> >
> > Thank the authors for addressing my concerns. I will maintain my rating.

---

> > > ### Author Response · Authors · 2025-11-25
> > >
> > > Thank you for your thoughtful feedback and for dedicating your time to review our submission. We are grateful for your positive assessment.

---

### Official Review · Reviewer_zaRL · 2025-11-01

**Soundness:** 3
**Presentation:** 2
**Contribution:** 2
**Rating:** 4
**Confidence:** 3

**Summary:**

This paper proposes BiCL, a framework for the continual adaptation of small language model-based embodied agents. The goal is to efficiently adapt to new tasks using only few-shot demonstrations and a small LM. The method consists of two main components: (1) "bidirectional CoT learning," which jointly trains the model to generate CoT rationales and to reflexivel correct rationales from a previously learned policy, and (2) "rationale-wise test-time scaling", a mechanism to dynamically stop the reasoning process at inference. The authors claim this bidirectional training creates an "asymmetric effect," where the model internalizes self-correction capabilities, leading to more robust single-pass inference without needing an explicit reflection step.

**Strengths:**

- The paper addresses a significant and practical challenge: how to continually adapt LM agents in resource-constrained environments where standard "pretrain-then-finetune" or large-scale prompting is not feasible.
- Within its chosen setting, the BiCL framework demonstrates consistently superior performance over relevant continual learning baselines, such as sequential finetuning (SeqFT-Distill) and adapter-based methods (TAIL-Distill).

**Weaknesses:**

- BiCL is not a general-purpose post-training technique. It is a complex system with many moving parts, each tailored to this specific embodied agent setup. This complexity makes the method difficult to reproduce and adapt to other domains, feeling less like a principled learning framework and more like a collection of specialized heuristics.
- The TTS mechanism relies on a threshold $\delta_k$ computed via a hyperparameter $\lambda$. The paper reports using $\lambda = -0.524$ for VirtualHome and $\lambda = 0.0$ for ALFWorld. A mechanism that requires environment-specific hyperparameter tuning is not robust. Furthermore, the sensitivity analysis shows that performance is indeed highly sensitive to this threshold, making it a fragile component rather than a stable efficiency gain.
- The paper treats reasoning (generation) and reflection (correction) as two distinct tasks. However, more general and powerful reasoning frameworks tend to integrate these capabilities. Effective reasoning often involves a single, unified process where the model generates thoughts, identifies potential flaws, and self-corrects within one continuous "long CoT" generation.

**Questions:**

Please refer to my weakness

---

> ### Author Response · Authors · 2025-11-19
>
> **`[Weakness 1] BiCL is not a general-purpose post-training technique. It is a complex system with many moving parts, each tailored to this specific embodied agent setup`**
>
> Our work specifically targets embodied agent settings, but the **design of BiCL itself is applicable to general sequential decision-making tasks formulated under the standard Markov Decision Process (MDP) setting**.
>
> The core components of BiCL, reasoning-policy and planning-policy, are derived from general principles for improving long-horizon decision quality, rather than from ad-hoc mechanisms tailored to a particular environment. These components naturally extend to any domain where decisions exhibit log-horizon dependencies matters, including sub-task decomposition, navigation, and procedural planning. Moreover, **BiCL builds upon the reasoning-planning architecture that is widely adopted in prior work [1-4], and its self-correction paradigm is a general strategy for improving multi-step reasoning quality, not a heuristic unique to embodied agents.** Therefore, although our empirical evaluation focuses on embodied-agent domains, the underlying methodology is general and readily adaptable to other MDP-based tasks.
>
> [1] Shinn, Noah, et al. "Reflexion: Language agents with verbal reinforcement learning." NeurIPS 2023
>
> [2] Zawalski, Michał, et al. "Robotic control via embodied chain-of-thought reasoning." CoRL 2024
>
> [3] Huang, Wenlong, et al. "Inner monologue: Embodied reasoning through planning with language models." CoRL 2022
>
> [4] Wen, Junjie, et al. "Diffusion-VLA: Generalizable and Interpretable Robot Foundation Model via Self-Generated Reasoning." ICML 2025
>
> ----
>
> **`[Weakness 2] TTS  requires environment-specific hyperparameter tuning is not robust. Furthermore, the sensitivity analysis shows that performance is indeed highly sensitive to this threshold.`**
>
> The threshold $\delta_k$ is **derived from the distribution of confidence scores using a percentile-based statistical rule**. Specifically, for each CoT step $k$, we compute the mean and variance of the log-probabilities of the ground-truth actions and convert a desired percentile cutoff into a z-score. Therefore, the value $\lambda$ is simply determined by the z-value associated with the chosen percentile (e.g., $\lambda = -0.524$ for the 30th percentile, and $\lambda = 0.0$ for the 50th percentile).
>
> Importantly, the **sensitivity analysis shows that BiCL remains robust across a wide range of thesholds**. Even when the least favorable threshold (Very High) is used, the performance drop relative to the best-performing threshold (Low) is modest, showing a SR drop of 6.26\% on the seen category and 5.84\% on unseen category. Moreover, when comparing Table 19 with baselines in Table 1, **BiCL consistently outperforms all baselines across all threshold settings while retaining its computational efficiency**. Notably, even under the worst-performing threshold (Very High), BiCL still exceeds the strongest baseline, SeqFT-Distill, by 9.37\% on the seen category and 12.41\% on the unseen category.
>
> We have added detailed explanations of the statistical threshold selection strategy in Appendix C.9 and expanded the sensitivity analysis in Appendix D.7 to further support these claims in the revised manuscript.
>
> ----

---

> ### Author Response · Authors · 2025-11-19
>
> **`[Weakness 3] The paper treats reasoning (generation) and reflection (correction) as two distinct tasks. However, more general and powerful reasoning frameworks tend to integrate these capabilities within one continuous "long CoT" generation.`**
>
> While unified long-form CoT generation that interleaves reasoning, flaw-identification, and self-correction is one possible approach, our objective of BiCL is fundamentally different. **Rather than requiring explicit reflection during inference, BiCL aims to internalize self-correction capability during training so that the model performs strong CoT reasoning without ant additional reflection at test time.** This distinction is especially important in embodied-agent settings, where the agent must operate under computational constraints.
>
> Moreover, the unified long-CoT paradigm presents two inherent limitations in embodied agent settings. First, performing self-correction within a **long CoT requires generating additional critique and revision tokens, which substantially increases inference-time compute** (as shown in Table 3). Second, **explicit self-correction depends on reliable feedback signals to identify and correct reasoning errors**. However, during inference, no ground-truth feedback is available, forcing long-CoT approaches to rely on self-generated critiques that may inherently noisy and unreliable, particularly for small LMs.
>
> To further highlight these limitation, we include a Self-Correction baseline in Table 3, which performs CoT reasoning followed by explicit reflection. As shown, explicitly executing reflection within a long CoT sequence yields substantially lower performance than BiCL, precisely because reliable external feedback is unavailable at inference time. These results demonstrate that, **in sLM-based embodied-agent settings, integrating reflection into a single long CoT sequence is not necessarily more powerful in terms of either reasoning quality and computational efficiency.** In contrast, BiCL's training-time internalization of reflexive reasoning leads to more stable, efficient, and robust reasoning at test-time.
>
> ----
>
> **`Summary Response`**
>
> We sincerely thank the reviewers for their insightful and constructive feedback. We have carefully addressed all comments and incorporated detailed clarifications. If there are any further questions or suggestions, we would be very glad to provide additional explanations.

---

### Official Review · Reviewer_K8gT · 2025-11-01

**Soundness:** 2
**Presentation:** 2
**Contribution:** 2
**Rating:** 4
**Confidence:** 3

**Summary:**

### Summary: Asymmetric Effects of Self-Corrective Learning

#### Research Problem
The paper addresses the challenge of **efficiently adapting Large Language Model (LLM)-powered embodied agents** to a stream of diverse, sequential tasks, especially under limited supervision (few-shot or zero-shot scenarios) and resource constraints. The goal is to maximize **forward transfer** (improving future tasks) while minimizing **catastrophic forgetting** (losing knowledge of past tasks).

#### Methodology (BiCL)
The authors propose **BiCL (Bidirectional Contrastive Learning)**, an embodied task adaptation framework that leverages Chain-of-Thought (CoT) reasoning.

1.  **Self-Corrective Learning:** BiCL uses a self-corrective mechanism to refine **both the CoT rationale and the corresponding action policy**. This is achieved by generating high-quality self-correction samples for both components.
2.  **Asymmetric Contrastive Learning (ACL):** ACL is introduced to guide the adaptation process:
    * **Forward Transfer (CoT Rationale):** ACL maximizes the **similarity** between the current task's rationale and those of *future* tasks.
    * **Catastrophic Forgetting (Action Policy):** ACL maximizes the **dissimilarity** between the current task's policy and those of *past* tasks. This asymmetric objective ensures the CoT is broad and transferable, while the policy remains discriminative and task-specific.
3.  **Rationale-Wise Test-Time Scaling:** A mechanism is used during inference to adapt the CoT's influence based on its confidence, promoting a **balanced trade-off** between reasoning and action selection.

#### Key Experiments
BiCL is evaluated on two complex embodied continual learning benchmarks:
* **ALFWorld** (textual environment).
* **VirtualHome Beh-IL** (visual environment with hierarchical actions).

**Results** show that BiCL significantly **outperforms state-of-the-art baselines** in both seen and unseen task categories, demonstrating superior **bidirectional adaptation** by achieving high forward transfer and minimal catastrophic forgetting.

**Strengths:**

### Strengths of the Paper (Asymmetric Effects of Self-Corrective Learning)

1.  **Novel Asymmetric Contrastive Learning (ACL) for Continual Adaptation:** The core strength is the **BiCL** framework's use of Asymmetric Contrastive Learning. It cleverly separates the learning objectives for the reasoning component (CoT rationale) and the action component (policy). It promotes **forward transfer** by making the CoT rationale broadly similar to future tasks, while simultaneously mitigating **catastrophic forgetting** by making the current policy discriminatively dissimilar from past policies. This bidirectional optimization is highly effective for continual learning.

2.  **Integrated Self-Correction and Dual Refinement:** The methodology incorporates a robust **self-corrective learning** mechanism that refines *both* the Chain-of-Thought (CoT) rationale and the *action policy* based on generated high-quality samples. This dual refinement ensures that the agent's internal reasoning process and its external physical actions are consistently optimized, leading to more reliable and accurate policy adaptation under limited supervision.

**Weaknesses:**

### Potential Weaknesses of the Paper (Asymmetric Effects of Self-Corrective Learning)

1.  **Complexity and Computational Cost of BiCL:** The proposed **BiCL** framework is inherently complex, involving three interconnected components: Chain-of-Thought (CoT) generation, self-correction for both rationale and policy, and Asymmetric Contrastive Learning (ACL). This complexity requires multiple forward and backward passes, likely leading to **significant computational overhead** compared to simpler fine-tuning or rehearsal methods, which may restrict its scalability or deployment in resource-constrained settings.

2.  **Sensitivity to Hyperparameters (Rationale-Wise Scaling):** The paper introduces a **Rationale-Wise Test-Time Scaling** mechanism, which relies on a threshold ($\delta_k$) to decide when to stop CoT reasoning. The results indicate that the performance is highly **sensitive to this threshold**, with both setting it too low or too high causing significant performance drops. This hyperparameter requires careful tuning per task or domain, potentially limiting its robustness and automatic deployment.

3.  **Dependence on High-Quality Self-Correction Samples:** The effectiveness of the entire approach hinges on the ability of the self-corrective mechanism to generate **high-quality, accurate self-correction samples** for both the CoT rationale and the action policy. If the initial LLM-powered agent generates poor self-corrections (e.g., due to insufficient initial knowledge or ambiguous environments), the learning process could be misguided, leading to error amplification rather than improvement.

4.  **Simulated Environment Bias:** While evaluated on two distinct benchmarks (**ALFWorld** and **VirtualHome Beh-IL**), both are **simulated environments**. The observed benefits in mitigating catastrophic forgetting and maximizing forward transfer might not translate seamlessly to the complexity, noise, and unpredictability of **real-world physical robot tasks**, where state observation is often partial and actions are subject to physical uncertainty.

**Questions:**

### Related Questions and Suggestions for the Paper

#### 1. Experimental Setup and Baselines (ACL vs. Self-Correction)

**Question:** Given that the BiCL framework combines **Self-Correction** and **Asymmetric Contrastive Learning (ACL)**, have the individual contributions and interplay of these two complex mechanisms been sufficiently isolated and analyzed?

**Suggestion/Query:**
* Could the authors provide a more detailed **Ablation Study** to compare the effects of: (a) Self-Corrective Learning only (without ACL), versus (b) ACL only (without self-correction)?
* This would help validate the specific role of ACL in promoting **forward transfer** and mitigating **catastrophic forgetting**, and confirm the method's superiority is not predominantly driven by the robust self-correction mechanism alone.

#### 2. Methodological Rationale (Asymmetry Justification)

**Question:** What is the fundamental **theoretical rationale** for the asymmetric design where the **Chain-of-Thought (CoT) rationale** aims to **maximize similarity** to future tasks, while the **Action Policy** aims to **maximize dissimilarity** from past tasks?

**Suggestion/Query:**
* Has the paper explored a **symmetric baseline** (e.g., maximizing similarity for both components) to rigorously demonstrate the necessity of this asymmetry?
* Furthermore, how does this mechanism reliably handle tasks with highly similar CoT steps (e.g., planning to move to the same location) but requiring distinct physical policies (e.g., "pick up" vs. "put down") without the maximizing dissimilarity objective causing destructive interference?

#### 3. Additional Experimental Content (Scale and Efficiency)

**Question:** Since BiCL targets **embodied continual learning**, do the current experiments fully cover the challenges of long-term learning and real-world deployment complexity?

**Suggestion/Query:**
* **Scale and Duration:** Can the **length of the task stream** and the **diversity of tasks** be significantly scaled up (e.g., by an order of magnitude) to provide a more stringent test of the model's robustness and long-term **forgetting rate**?
* **Efficiency Metrics:** As BiCL is computationally complex, it would be beneficial to include a detailed analysis of the **training time and inference latency** overhead introduced by the self-correction and ACL components, establishing the method's **efficiency-performance trade-off** compared to simpler baselines.

---

> ### Author Response · Authors · 2025-11-21
>
> **`[Overall Response]`**
>
> After carefully reading all parts of the review, we found that **substantial statements from the reviewer do not correspond any part of our paper**. We conjecture that these inconsistencies appear to arise from a severe misunderstanding about the core components of our framework. We list the major inaccuracies and clarify the key points below.
>
> * **Bidirectional Contrastive Learning (BiCL)** : The acronym BiCL stands for Bidirectional CoT Learning, not Bidirectional Contrastive Learning. Furthermore, our framework does not include any form of contrastive learning objective.
>
> * **Asymmetric Contrastive Learning (ACL)** : The paper does not propose or mention any form of contrastive learning or asymmetric objective. No contrastive objective is defined or optimized anywhere in the paper.
>
> * **Similarity maximization between current task's rationale and those of future tasks**: The method does not optimize rationale similarity with future tasks. Similarity is used only once for adapter selection as a retrieval heuristic, and it is never part of the training objective.
>
> * **Dissimilarity maximization for action policies**: No such mechanism exists. More importantly, the paper does not even use the term action policy at all. Our method employs a planning-policy, which is trained solely through supervised action reconstruction and does not involve any dissimilarity-based objective.
>
> As most of the reviewer’s weaknesses and questions stem from misunderstandings of the core components of our method, it becomes challenging to provide fully meaningful or constructive responses. Nevertheless, we address each point individually below and clarify the intended design and contributions of our framework.

---

> ### Author Response · Authors · 2025-11-21
>
> **`[Weakness 1] Complexity and Computational Cost of BiCL`**
>
> First, **our framework does not employ any form of contrastive learning, and therefore asymmetric contrastive learning (ACL) is not part of our method**. Our BiCL framework consists of two core components: (i) a reasoning-policy responsible for generating rationale segments, and (ii) a planning-policy that predicts actions conditioned on these rationales. The self-correction process applies only to the reasoning-policy during training, where the model learns to refine previously generated rationales. Importantly, no self-correction is performed for the planning-policy. Moreover, BiCL introduces only a single additional reflexive reasoning loss on top of the standard CoT reasoning objective. This loss is computed using the same forward pass of the reasoning-policy, with negligible extra computation beyond a lightweight embedding similarity check. Since BiCL does not introduce auxiliary networks, contrastive objectives, or extra backward passes, the overall training complexity remains comparable to standard LoRA fine-tuning.
>
> ----
>
> **`[Weakness 2] Sensitivity to Hyperparameters (Rationale-Wise Scaling)`**
>
> **The claim that performance is "highly sensitive" to $\delta_k$ does not align with our empirical results**. Importantly, the sensitivity analysis shows that BiCL remains robust across a wide range of thesholds. Even when the least favorable threshold (Very High) is used, the performance drop relative to the best-performing threshold (Low) is modest, showing a SR drop of 6.26\% on the seen category and 5.84\% on unseen category. Moreover, when comparing Table 19 with baselines in Table 1, BiCL consistently outperforms all baselines across all theshold settings while retaining its computational efficiency. Notably, even under the worst-performing threshold (Very High), BiCL still exceeds the strongest baseline, SeqFT-Distill, by 9.37\% on the seen category and 12.41\% on the unseen category.
>
> We have added detailed explanations of the statistical threshold selection strategy in Appendix C.9 and expanded the sensitivity analysis in Appendix D.7 to further support these claims in the revised manuscript.
>
> ----
>
> **`[Weakness 3] Dependence on High-Quality Self-Correction Samples`**
>
> To ensure the high quality of rationales produced by the LLM, we employ a two-phase annotation pipeline. In the first stage, we prompt the LLM to generate multiple rationale candidates for each predefined query set. In the subsequent stage, we select the rationale set that yields the highest log-probability of the ground-truth future plan when used as input to the planning-policy. The full rationale annotation pipeline is detailed in Appendix C.1.
>
> Through this process, **all annotated rationales are grounded directly in expert demonstrations and filtered by the model’s own action likelihood**, ensuring that the rationales are coherent and aligned with behaviors that maximize task success. This mitigates the risk of low-quality or incorrect rationales and provides a stable supervision signal for distilling high-quality reasoning from LLMs into small LMs.
>
> ----
>
> **`[Weakness 4]  Simulated Environment Bias`**
>
> To better reflect the real-world conditions, **we conduct additional experiments under partial perception by independently dropping each triple in state** with probabilities of 5\% and 10\% (0\% drop corresponds to the default setting), simulating intermittent sensory failures. As shown in the below table, the performance drop is negligible, showing only SR drop of 3.87\% for the seen category and 1.20\% for the unseen category under 10\% drop setting. This is attributed to **our robust CoT reasoning which enables the model to maintain resilience from missing perception, demonstrating its suitability for real-world deployment**.
>
> | Method           | Seen SR (%)            | Seen GC (%)            | Unseen SR (%)          | Unseen GC (%)          |
> |------------------|-------------------------|-------------------------|-------------------------|-------------------------|
> | (0% drop) L2SC   | 78.87 ± 1.76            | 85.25 ± 1.29            | 62.09 ± 2.30            | 69.88 ± 1.98            |
> | (5% drop) L2SC   | 76.62 ± 1.90 (−2.25)    | 84.08 ± 1.37 (−1.17)    | 61.70 ± 2.28 (−0.39)    | 69.88 ± 1.96 (−0.00)    |
> | (10% drop) L2SC  | 75.00 ± 1.97 (−3.87)    | 83.07 ± 1.44 (−2.18)    | 60.89 ± 2.29 (−1.20)    | 69.64 ± 1.97 (−0.24)    |

---

> ### Author Response · Authors · 2025-11-21
>
> **`[Question 1] Given that the BiCL framework combines Self-Correction and Asymmetric Contrastive Learning (ACL), have the individual contributions and interplay of these two complex mechanisms been sufficiently isolated and analyzed?`**
>
> As mentioned in the Overall response and Weakness 1, **there is no Asymmetric Contrastive Learning in our framework**. BiCL consists only of: (i) Bidirectional CoT learning, which jointly optimizes CoT reasoning and reflexive reasoning using demonstration rationales; and (ii) Rationale-wise test-time scaling, which adaptively determines the reasoning depth based on the planning-policy’s confidence. The ablation of bidirectional CoT learning and rationale-wise test-time scaling is ablated in Table 5 and Table 3, respectively.
>
> ----
>
> **`[Question 2] What is the fundamental theoretical rationale for the asymmetric design where the Chain-of-Thought (CoT) rationale aims to maximize similarity to future tasks, while the Action Policy aims to maximize dissimilarity from past tasks?`**
>
> As mentioned in the Overall Response and Weakness 1, **BiCL does maximize similarity or dissimilarity across tasks for either the CoT reasoning-policy or the action planning-policy. These objectives are not part of our framework**.
>
> ----
>
> **`[Question 3] Since BiCL targets embodied continual learning, do the current experiments fully cover the challenges of long-term learning and real-world deployment complexity?`**
>
> While Table 1 reports results under a 4-stage continual adaptation setting, **Table 2 evaluates BiCL under a 7-stage continual adaptation setting, showing the scalability of BiCL in terms of long-term adaptation stage**.  We inadvertently omitted this clarification in the main text. In the revised manuscript, we have added a sentence to each table's descriptive paragraph to explicitly state the corresponding number of adaptation stages.
>
> Moreover, in Appendix D.5, we provide experiments on long-horizon compositional tasks in VirtualHome, demonstrating that BiCL maintains strong performance even as task complexity increases. Furthermore, to approximate real-world deployment challenges under imperfect perception, we conduct robustness experiments in Weakness 4.
>
> ----
>
> **`[Summary Response]`**
>
> We sincerely thank the reviewer for the time and effort dedicated to evaluating our submission. We carefully reviewed all comments, and we believe that **many of the reviewer's concerns raised stem from misunderstandings about the core components of our method**. In particular, several weaknesses and questions refer to mechanisms that are not part of our framework and do not appear anywhere in the manuscript (e.g., asymmetric contrastive learning, action-policy self-correction, or similarity/dissimilarity objectives).
>
> Given that many of the critiques are based on elements not present in our paper, **we kindly ask the reviewer to re-evaluate the submission in light of our clarifications**. We appreciate your consideration and thank you again for your thoughtful review.

---

### Meta-Review · Area_Chair_fX5e · 2026-01-04

**Summary:**

Most of the reviews are not positive and they raised many issues related to significance, novelty and evaluation. The author response addressed some of concerns in some way. But overall, the quality of the paper seems below the bar of ICLR.

**Reviewer Concerns:**

Some concerns are not fully addressed such as " This paper situates its method within a very limited scope of existing literature on task adaptation for embodied agents. In particular, RL is a standard approach for enabling LLM-based agents to adapt across tasks without supervised data, yet it is not addressed in either the related work or the baselines." by reviewer 6u9M.

**Reviewer Scores:**

Most reviewers will slightly increase their scores.

---

### Decision · Program_Chairs · 2026-01-26

Reject